# Interconnected nanoconfining pore networks enhance catalyst $CO_2$ interaction in electrified reactive capture

Hengzhou Liu[1,8], Lun An[2,8], Peiyao Wang[3,8], Christine Yu[1,8], Jie Zhang[2], Heejong Shin[1], Bosi Peng[1], Jiantao Li[1], Matthew Li[4], Hongmin An[1], Jiaqi Yu[1], Yuanjun Chen[1], Peiying Wang[1], Kug-Seung Lee[5], Kanika Lalit[6], Zeyan Liu[1], Omar K. Farha[1], Wenyu Huang[2,6], Jefferson Zhe Liu[3] ✉, Long Qi[2] ✉, Ke Xie[1] ✉ & Edward H. Sargent[1,7] ✉

Systems that sequentially capture and upgrade $CO_2$ from air to fuels/fuel-intermediates, such as syngas and ethylene, rely on an energy-intensive $CO_2$ release process. Electrified reactive capture systems transform $CO_2$ obtained directly from carbonate capture liquid into products. Previous reactive capture systems show a decline in Faradaic efficiencies (FE) at current densities above 200 mA/cm². Here we show the chemical origins of this problem, finding that prior electrocatalyst designs failed to arrest, activate, and reduce in situ-generated $CO_2$ ($i$-$CO_2$) before it traversed the catalyst layer and entered the tailgas stream. We develop a templated synthesis to define pore structures and the sites of Ni single atoms, and find that carbon-nitrogen-based nanopores are effective in accumulating $i$-$CO_2$ via short-range, non-electrostatic interactions between $CO_2$ molecules and the nanochannel walls. These interactions confine and enrich $i$-$CO_2$ within the pores, enhancing its binding and activation. We report as a result carbonate electrolysis at 300 mA/cm² with FE to CO of 50% ± 3%, and with <1% $CO_2$ in the tailgas outlet stream. This corresponds to a projected energy efficiency (EE) to 2:1 syngas of 46% at 300 mA/cm² when $H_2$ is added using a water electrolyzer.

Carbon capture and utilization (CCU)—such as direct air capture (DAC) followed by electrochemical upgrade to fuels/fuel precursors—offers, if energy-efficient and powered using low-carbon electricity, to lower the carbon intensity of fuels and chemicals (Fig. 1a)[1–5]. For the DAC stage, an alkaline solution such as KOH captures $CO_2$ as $K_2CO_3$[5], and concentrated $CO_2$ is released through energy-intensive drying and calcination steps at ~900 °C, requiring 8–10 GJ/tonne $CO_2$. Considering

the case of upgrade of this $CO_2$ (44 g/mol) to CO (28 g/mol), this adds a further energy consumption of $(44/28) \times (8\text{-}10 \text{ GJ/tonne } CO_2) = 13\text{-}16$ GJ/tonneCO. This value, which is additive atop the $CO_2$-to-CO electrolyzer energy, is appreciable, itself residing well above the LHV of the final intended product.

Reactive capture, by contrast, integrates $CO_2$ capture and conversion in a single system (Fig. 1b)[6–10]. In comparison with the low $CO_2$

¹Department of Chemistry, Northwestern University, Evanston, IL, USA. ²U.S. DOE Ames National Laboratory, Iowa State University, Ames, IA, USA. ³Department of Chemical Engineering, The University of Melbourne, Parkville, VIC, Australia. ⁴Chemical Sciences and Engineering Division, Argonne National Laboratory, Lemont, IL, USA. ⁵Pohang Accelerator Laboratory (PAL), Pohang University of Science and Technology (POSTECH), Pohang, Republic of Korea. ⁶Department of Chemistry, Iowa State University, Ames, IA, USA. ⁷Department of Electrical and Computer Engineering, Northwestern University, Evanston, IL, USA. ⁸These authors contributed equally: Hengzhou Liu, Lun An, Peiyao Wang, Christine Yu. ✉e-mail: zhe.liu@unimelb.edu.au; lqi@iastate.edu; ke.xie@northwestern.edu; ted.sargent@northwestern.edu

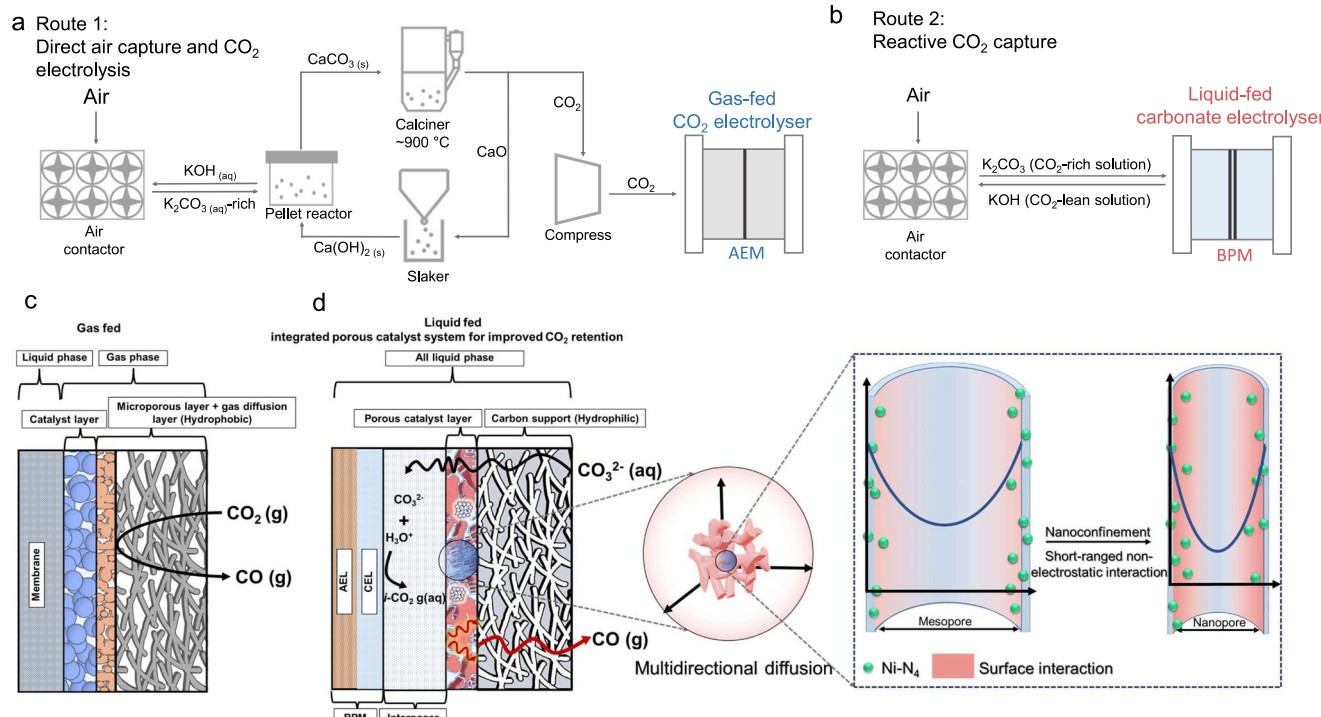

**Fig. 1 | Schematic illustration of systems. a** Sequential process of direct air capture (DAC) coupled with $CO_2$ electrolysis (Route 1) vs. **b** integrated process of reactive $CO_2$ capture (Route 2). **c, d** Comparison of $CO_2$ transport and conversion in two systems: **c** A gas-fed $CO_2$ electrolysis system incorporating a hydrophobic microporous layer, a gas diffusion layer, and a catalyst layer. The $CO_2$ reduction reaction took place at the catalyst surface within a three-phase boundary. **d** The current carbonate-based reactive capture system operates in a hydrophilic environment. In this work, we integrate a porous catalyst supported on carbon, featuring multidirectional nanoporous diffusion channels. These structures promote short-range, non-electrostatic interactions between pore walls and reactants, enhancing the retention and accumulation of $i$-$CO_2$ and thereby maximizing its utilization. Source data for the results are provided as a Source Data file.

concentration in air (~400 ppm), a post-capture liquid consisting of carbonate-rich solution[5,11] provides a more concentrated carbon-containing feedstock. Both bipolar membrane (BPM) and cation exchange membrane (CEM)-based electrolyzers have been advanced in carbonate-liquid-fed system[12–15].

In conventional, gas-fed, $CO_2$ reduction systems (Fig. 1c), a hydrophobic support is employed, and $CO_2$ gas-phase transport occurs through the gas diffusion layer and a microporous layer, providing an abundance of the needed feedstock to the catalyst surface[16–19].

In contrast, carbonate-liquid-fed systems (Fig. 1d and Supplementary Fig. 1) are flooded, and $CO_{2(g(aq))}$, i.e., gaseous $CO_2$ dissolved in aqueous solution, is generated only once $CO_3^{2-}$(aq) reaches the cation exchange layer (CEL) side of the BPM and is acidified via $H_3O^+$ provided by water dissociation in the BPM. The $CO_{2(g(aq))}$ must then travel back to the catalyst, moving first through a ~100 μm thick porous interposer (its purpose to establish a pH difference between CEL and catalyst: acidic at the CEL edge for $CO_2$ regeneration, near-alkaline at the catalyst to avoid undue HER)[14]. The rate of provisioning of this $i$-$CO_2$ is limiting in reactive capture systems (Supplementary Note 1): at current density 100 mA/cm², the best-case $i$-$CO_2$ supply rate is ~0.7 mL min⁻¹ cm⁻²; the value is lower during electrolysis, during which OH⁻ generation leads to an even higher local pH that further consumed $i$-$CO_2$. Thus, successful reactive capture systems must be maximally selective and active for $CO_2$ reduction, even in an $i$-$CO_2$-starved environment[14].

The cathode catalyst in prior reactive capture systems has typically been a substantially planar catalyst (Supplementary Fig. 2), such as Ag (for CO) and Cu (for $C_{2+}$ products)[12,13,15,20–22], the layer residing atop the flooded hydrophilic support. We noted that such systems diminish in their performance (e.g., the FE to reduced carbon products

declines below 30%) even at moderate current densities such as 100 mA/cm² and above. Similar trends have been seen in reactive capture using Ni single-atom[21] and molecular[14,23] catalysts.

## Results

### Nickel single-atom catalysts with engineered pore structure

We hypothesized that, in carbonate-fed reactive capture systems, a porous catalyst (Fig. 1d) could—once optimal conditions of pore size and pore density were found—facilitate $CO_2$ retention and accumulation at the 3D solid-phase dispersion of electrocatalytic sites[24]. We therefore adopted an approach in which the porous transport layer itself served as the catalyst. Once $i$-$CO_2$ was delivered, it would then locally bind and activate to provide $CO_2$ reduction. We posited that engineering of (1) the degree of pore alignment vs. anisotropy (2) the size of pores could maximize the $i$-$CO_2$:catalyst interaction, enabling selectivity in favor of $CO_2$ electroreduction even as current density increased.

Since Ni single atoms (Ni-SAC) are known-good $CO_2$-to-CO catalysts, we sought to form this porous catalyst from Ni, coordinated with nitrogen-carbon matrix, in turn dispersed and coordinated into a conductive carbon support. We airbrushed the resulting catalyst onto a hydrophilic substrate made of carbon paper. Porous Ni single-atom catalysts are prepared through a template-controlled coordination-condensation-carbonization synthesis (Supplementary Fig. 3 and "Methods")[25]. $Ni^{2+}$ is first coordinated with ethylenediamine and polymerized with carbon tetrachloride within the pores of silica template, after which the template is removed. This synthesis process enables the pre-coordination of Ni with N, improves Ni dispersion, and allows precise control over the morphology and porous structure.

We varied the pore structure of candidate porous catalysts by using a library of templates that modulated both the pore channel

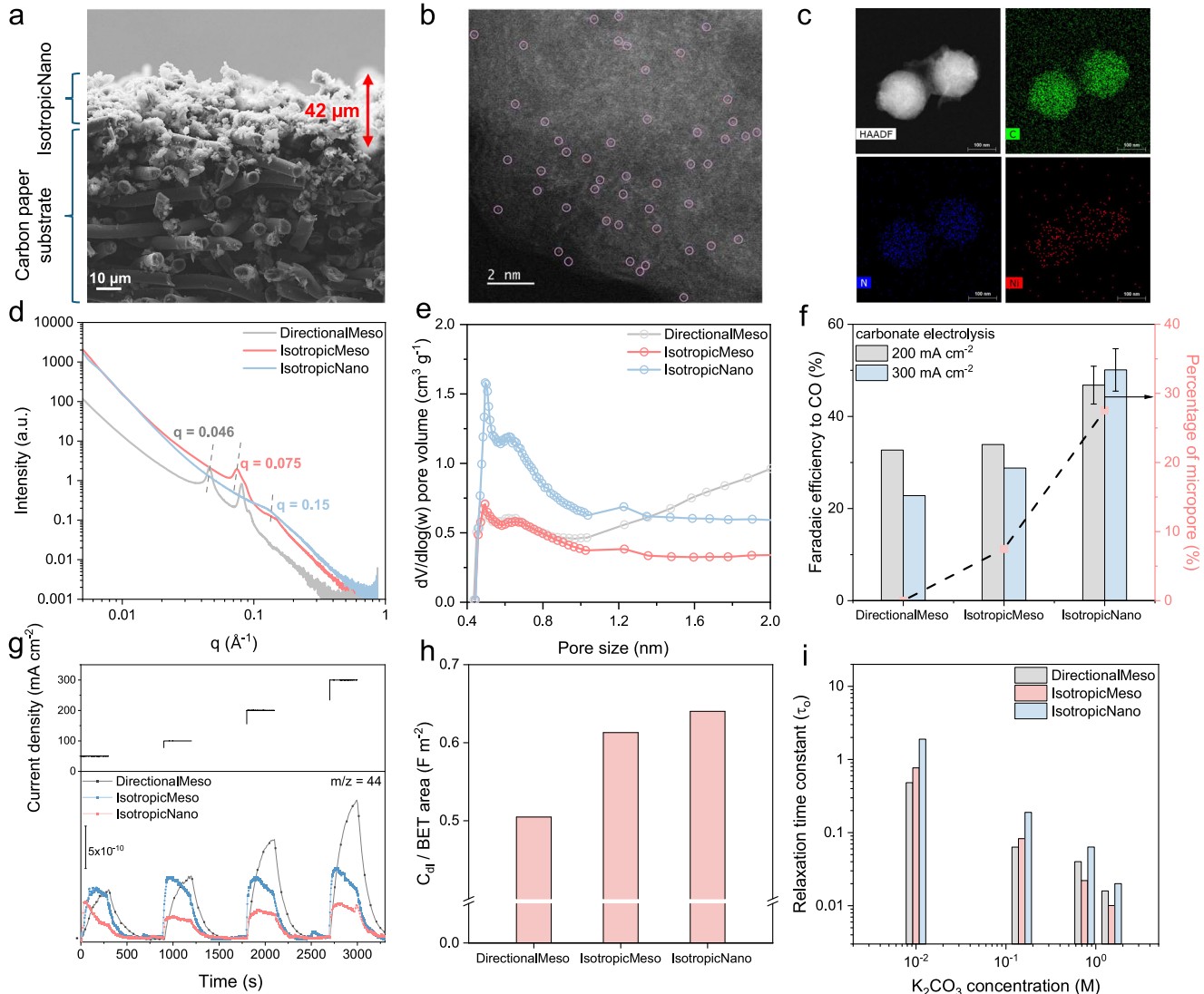

**Fig. 2 | *i*-CO₂ transport kinetics in pores. a** Cross-sectional SEM image of the cathode electrode consisting of the IsotropicNano catalyst coated on a hydrophilic carbon paper substrate. **b** HRTEM image and **c** HAADF-STEM-EDS mapping of the IsotropicNano catalyst. **d** Synchrotron-based Small-Angle X-ray Scattering (SAXS) profiles and **e** BET pore size distribution in the 0.4–2 nm range. **f** Faradaic efficiency (FE) to CO and percentage of nanopores as a function of cathode electrode at 200 and 300 mA/cm² in a carbonate electrolyzer with BPM. Where error bars are shown, values are means, and error bars indicate s.d. ($n = 3$ replicates). **g** EC-MS signal of CO₂ ($m/z = 44$) measured across a current density range of 50–300 mA/cm². **h** Total capacitance ($C_{dl}$) normalized by the BET surface area. **i** Calculated $\tau_o$ in dilute K₂CO₃ electrolytes. Relaxation time constant ($\tau_o$) is calculated from the peak frequency ($f_o$) of imaginary part of complex capacitance ($C_{im}$) using the relation $\tau_o = (2\pi f_o)^{-1}$. Source data for the results are provided as a Source Data file.

topology and the average pore diameter. We explored three templates (Supplementary Fig. 4 and Supplementary Table 1) and also varied the post-synthesis NH₃ treatment. One class of template, whose corresponding porous catalyst we term DirectionalMeso, is expected to provide substantially 1D transport through the catalyst layer. A second, IsotropicMeso, consists of mesopores (2–50 nm) having no directional preference. A third, IsotropicNano, includes both meso and nano (<2 nm) pores. In each case, we optimized the porous catalysts thickness, finding that a 40 μm (Fig. 2a) thickness maximized electrolysis performance.

Each class of porous catalyst consisted of similar graphitic carbon (XRD and HRTEM, Supplementary Figs. 5–13). XPS (Supplementary Fig. 14) showed in each case N 1*s* spectra of four types of nitrogen (pyridinic, pyrrolic, graphitic, and oxidized N) in similar ratios. Elemental analysis showed similar nitrogen content (9–10 wt%, Supplementary Table 2). The Ni 2*p* spectra, with binding energies ~854.2 eV, indicated a slightly oxidized state of Ni[26]. STEM-HAADF imaging

(Fig. 2b and Supplementary Figs. 6–13) indicated the formation of single-atom Ni, while STEM-energy-dispersive X-ray spectroscopy (STEM-EDS) mapping suggested a uniform distribution of Ni within each porous catalyst (Fig. 2c).

Synchrotron-based Small-Angle X-ray Scattering (SAXS) indicated periodic and symmetric pore arrangements[27–29], as shown by correlation peaks in the profiles (Supplementary Note 2 and Fig. 15). Fractal analysis[29] showed mass fractal structures (fractal dimensions: D = 1–3), where the mass of the porous network scales non-linearly with size, consistent with self-similar structures formed via templating. Higher D values in IsotropicNano suggest increased spatial complexity, indicating greater surface roughness and a more disordered pore structure. The primary SAXS peaks (Fig. 2d), corresponding to different Miller indices, reflect average pore-to-pore distances that decrease from ~15 nm in DirectionalMeso to ~3 nm in IsotropicNano. It indicated IsotropicNano possesses the smallest average pore diameter, consistent with its high nanopore fraction

and the cubic lattice arrangement of its pore network (Supplementary Note 2).

$N_2$ adsorption-desorption isotherms further showed type-IV hysteresis loops indicative of a mesoporous structure (Supplementary Figs. 16–20 and Table S3). Brunauer–Emmett–Teller (BET) surface area and average pore size analysis indicated a predominance of mesopores in DirectionalMeso and IsotropicMeso, whereas IsotropicNano exhibited the expected higher proportion of nanopores (Supplementary Figs. 16–20 and Fig. 2e). SEM, TEM, and HAADF-STEM imaging revealed distinctive pore morphologies (Supplementary Figs. 6–13, 21–26): DirectionalMeso displayed a cylindrical structure with unidirectional channels and a linear pore array, whereas IsotropicMeso and IsotropicNano exhibited nanosheet and nanosphere structures.

### Electrolysis performance as a function of pore structure

We integrated the catalyst layers into a BPM-based system fed with carbonate solution. By optimizing the Ni precursor amount, calcination temperature, and adjusting the pore channel length within the 0.35–0.6 μm range, we achieved a peak $FE_{CO}$ of 40% at 100 mA/cm2 (Supplementary Fig. 27) on DirectionalMeso. However, this $FE_{CO}$ rapidly declined to below 30% at 200–300 mA/cm² (Fig. 2f).

In the IsotropicMeso catalyst, i.e., having no directional preference and a pore diameter ~ 2–50 nm, we observed a higher $FE_{CO}$ in the higher current density range of 200–300 mA/cm², though $FE_{CO}$ remained below 35% (Fig. 2f). It was only when we combined multi-directional diffusion with the addition of <2 nm nanopores in IsotropicNano that $FE_{CO}$ rose to 50% ± 3%, accompanied by the lowest extent of the $OH^-$–$CO_2$ neutralization side reaction and the lowest $CO_2$ gas evolution at the cathode outlet (Supplementary Table 4).

Inductively coupled plasma mass spectrometry (ICP-MS) analysis (Supplementary Table 5) indicates that the increased catalyst activity in IsotropicNano was not a result of higher Ni content, for IsotropicNano exhibited the lowest Ni mass loading (1.35 wt%) among all samples. The normalized Turnover Frequency (TOF) for CO generation in IsotropicNano was 1.9 and 3.3 times higher than that of DirectionalMeso and IsotropicMeso, respectively, at 300 mA/cm². Further increasing the Ni content in IsotropicNano did not enhance performance: it leads to nanoparticle formation, favoring the hydrogen evolution reaction (HER) (Supplementary Figs. 28–29).

Control experiments showed that, unlike hydrophobic gas-fed $CO_2RR$ systems or non-porous Ag-based catalysts—where only the surface layer actively participates in the reaction—an optimal IsotropicNano loading of 1.5–2 mg/cm² is essential in the fully-flooded carbonate system to maximize Ni single-site utilization throughout the entire catalyst layer (Supplementary Note 3 and Figs. 30–32).

We further employed in situ electrochemical mass spectrometry (EC-MS) to detect the unreacted $i$-$CO_2(g)$ at the cathode outlet (Fig. 2g). IsotropicNano showed minimized $CO_2(g)$ present in the tail-gas, especially at high current densities. In contrast, the one-dimensional/directional diffusion in DirectionalMeso may provide too-efficient egress of $i$-$CO_2$ from the entrance of the porous catalyst layer (facing the interposer and the CEM side of the BPM) to its plane of egress (atop the GDL), preventing selectivity in $i$-CO2RR from being retained at high currents. The 3D interconnected pore network in IsotropicNano—particularly with its higher fraction of nanopores—enhanced $i$-$CO_2$ retention and promoted its efficient utilization within the catalyst layer.

These results highlight that the entire catalyst layer plays a crucial role in facilitating $i$-$CO_2$ transport, retention, and penetration into the interconnected pores, thereby maximizing catalytic efficiency.

### Characterization of $i$-$CO_2$ transport kinetics in pores

We employed electrochemical impedance spectroscopy (EIS) and calculated the electric double-layer capacitance ($C_{dl}$), which represents the electrochemically-accessible surface area (electrolyte-wetted) for

the reaction (Supplementary Fig. 33)[30–32]. The BET area provides the physical surface area of the catalyst. Reporting $C_{dl}$ normalized to BET, $C_{dl}$/BET (Fig. 2h), reflects the fraction of the catalyst's physical surface area that actively participates in the electrochemical reaction, indicating active site utilization. In a 1.5 M $K_2CO_3$ electrolyte, the total capacitance of IsotropicNano (580 F g⁻¹) exceeds that of the other two catalysts (430 F g⁻¹ for DirectionalMeso and 490 F g⁻¹ for IsotropicMeso). IsotropicNano achieved the highest $C_{dl}$/BET of 0.64.

We then examined $CO_2$ diffusion and affinity under dry, wetted, and wetted with applied bias conditions. Under dry conditions, $CO_2$ affinity was assessed using $CO_2$ adsorption isotherm measurements. IsotropicNano showed increased $CO_2$ uptake over a wide pressure range (Supplementary Fig. 34). The measured $CO_2$ adsorption heat for IsotropicNano ranged from 24 to 36 kJ mol⁻¹, consistently higher than that of DirectionalMeso and IsotropicMeso.

Under wet conditions, we evaluated reactant retention within the pores, expressed by the relaxation time constant ($\tau_o$) (Fig. 2i) measured via EIS. $\tau_o$ reflects the rate of reactant transport in the pores, reporting thus on resistance to rapid penetration, and is sensitive to both the pore size and the pore tortuosity[32]. Figure 2i shows a trend in $\tau_o$: IsotropicNano (20 ms) > IsotropicMeso (16 ms) > DirectionalMeso (10 ms). In more dilute electrolytes, the $\tau_0$ differences among the catalysts increased further (Fig. 2i and supplementary Fig. 35).

We employed gas-fed assessed $CO_2$ electrolysis systems to study $CO_2$ transport and retention further (Supplementary Fig. 36). In three-compartment flow cells, IsotropicNano maintained a high FE of > 90% even as the $CO_2$ flow rate was turned down to 2.5 mL min⁻¹ cm⁻² and the $CO_2$ partial pressure was reduced to 20%. In a membrane electrode assembly (MEA)-based $CO_2$ electrolyzer with an anion exchange membrane (AEM), IsotropicNano achieved an impressive single-pass (SP) efficiency, the ratio of CO outlet flow to total $CO_2$ input flow, of 45% at 300 mA/cm² (Supplementary Fig. 37).

These experimental results indicate that the pore structure of IsotropicNano enhances the interaction of $CO_2$ with catalyst active sites, extending $i$-$CO_2$ retention within pores. This enables a greater accumulation of $i$-$CO_2$ within nanoconfined spaces.

### Simulation of $i$-$CO_2$ transport kinetics in pores

Our electrochemical reactive capture systems in confined pores involve tightly coupled processes—transport, adsorption, solvation, charge distribution, and reactions—spanning multiple scales[33]. These effects are strongly influenced by local pore environments and remain challenging to fully capture with current models, especially under nanoconfinement where continuum assumptions break down.

$CO_2$ adsorption and mobility within nanoporous environments influence reactant availability at electrochemical interfaces[33]. The physisorption of $CO_2$, in particular, plays a central role by establishing a local reservoir of molecules that are readily available to participate in subsequent, slower steps such as chemisorption and electron transfer[34,35]. Understanding how pore geometry and surface characteristics influence $CO_2$ adsorption and diffusion is of use for optimizing catalyst design and performance. In this context, our simulation work focuses specifically on the physisorption and transport behavior of $CO_2$ in sub-3 nm pores by combining density function theory (DFT) and molecular dynamics (MD) simulations.

We then performed DFT calculations to evaluate the $CO_2$ adsorption energy as a function of the distance normal to the catalyst surface (Fig. 3a). When the surface is uncharged, as the distance decreases from 9 Å to 3 Å, the $CO_2$ adsorption energy changes from zero (non-adsorption) to negative values (adsorption) and eventually to positive values (repulsion). We observed a maximum adsorption energy of −0.175 eV at a distance of 3.1 Å, where the O-C-O angle remains at 180° and aligns above the Ni atom (inset of Fig. 3a), indicating a physisorption interaction between $CO_2$ and uncharged Ni. Removing Ni atoms, i.e., changing to pure graphene, results in a lower

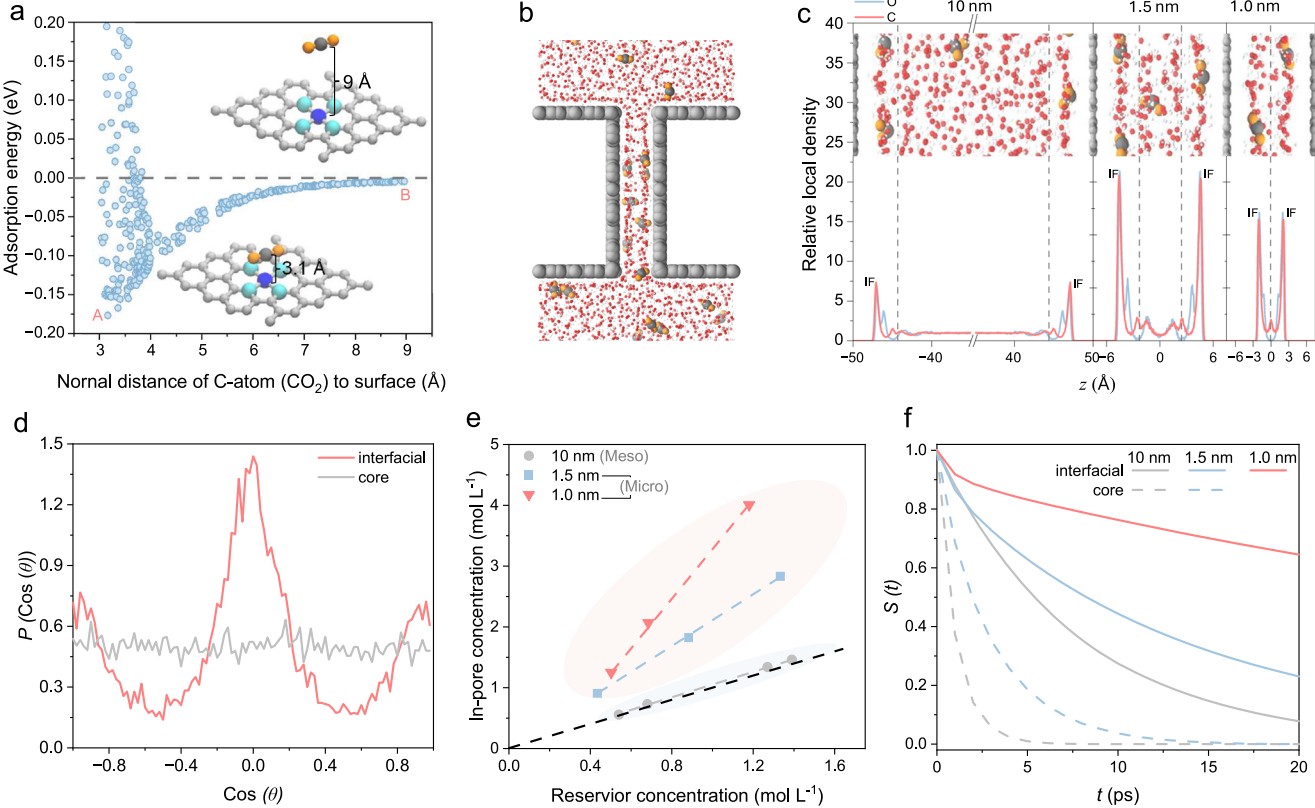

**Fig. 3 | Simulations of *i*-CO₂ transport kinetics in pores. a** $CO_2$ adsorption energy as a function of distance normal to the Ni single atom surface, calculated using density functional theory (DFT). A total of 400 cases was considered, accounting for $CO_2$' random rotations and distributions in the parallel plane at each specified normal distance. **b** Schematic showing the slit-based model employed in molecular dynamics (MD) simulations to approximate pore-size effects. **c** Local relative density distribution in the normal direction to the pore surface, comparing three pore sizes: 1.0 nm, 1.5 nm, and 10 nm. **d** The orientation analysis of $CO_2$ within the 1.0 nm slit. The angle $\theta$ is defined as the direction of C–O bond relative to the normal plane of the catalyst surface. **e** Summary of $CO_2$ in-pore concentration as a function of $CO_2$ reservoir concentration, comparing different pore sizes: 10 nm, 1.5 nm, and 1.0 nm. **f** The survival time correlation function ($S(\tau)$) as a function of time interval ($\tau$) within the interfacial and center regions of a nanoslit. Source data for the results are provided as a Source Data file.

$CO_2$ adsorption energy (i.e., 0.160 eV, Supplementary Fig. 38), indicating that the atomically dispersed Ni enhances $CO_2$ physisorption. When the Ni single atom surface becomes charged, recent work shows that the $CO_2$ adsorption would be further enhanced, where $CO_2$ is chemisorbed at the surface[36]. The $H_2O$ adsorption energy on Ni single atom surface is less than half that of CO, suggesting a means by which the rate of HER is diminished (Supplementary Fig. 39).

We employed MD to investigate the $CO_2$ distribution in a nanoconfined pore environment filled with water molecules. The MD model employs a nanoslit in contact with bulk reservoirs on both sides (Fig. 3b). We considered three slit sizes, 1.0 nm, 1.5 nm, and 10 nm, and varied $CO_2$ concentration in the bulk reservoir to represent dynamic $CO_2$ concentration in reactive capture systems operating at different current densities. We refined the force field parameters in MD simulations using a genetic algorithm (see Methods for details), training these to match DFT results.

The results show that reducing the slit size to sub-2 nm enhances $CO_2$ participation within the pore. When the slit surface is uncharged, and the slit size is 10 nm, the in-pore $CO_2$ concentration (red line) is equal to the bulk reservoir concentration (black line). As the slit size decreases to below 2.0 nm, the in-pore concentration surpasses the bulk concentration by 4x (Fig. 3e). This enhanced participation is a result of enriched $CO_2$ adsorption in the near-surface region, i.e., two sharp $CO_2$ density peaks near the surface in Fig. 3c, consistent with the DFT results (Fig. 3a). As slit size decreases, the pore volume occupied by these $CO_2$ layers increases (Supplementary Fig. 40), raising the overall $CO_2$ in-pore concentration. Our results show that the

nanoconfinement effect arises from short-range non-electrostatic interactions between the pore surface and $CO_2$ molecules, leading to increased reorientation and rearrangement of $CO_2$ within the pore (Fig. 3d and Supplementary Fig. 41). MD indicates that $H_2O$ does not interfere with $CO_2$ adsorption at the catalyst surface (Supplementary Fig. 42), consistent with its weaker surface adsorption energy.

The mobility/transport kinetics of $CO_2$ within confined pores were also investigated in MD simulations using the survival time correlation function, $S(\tau)$, which denotes the likelihood that $CO_2$ remains within a specified region over a time interval τ (Methods). We analyzed $S(\tau)$ for $CO_2$ in the interfacial (within 5.0 Å distance from the surface) and central regions of the nanoslit. In Fig. 3f, the interfacial region $S(\tau)$ decays much more slowly than the central region, indicating a significantly lower $CO_2$ mobility at the interface. In addition, the smaller pores have a higher interfacial $S(\tau)$ value, indicating that confinement further suppresses the transport dynamics of $CO_2$.

Our results offer insights into sub-nanometer variations in $CO_2$ adsorption density, interfacial residence times, and transport kinetics —key factors in determining reactant availability at active sites. They are complementary to conventional continuum models, such as Poisson–Nernst–Planck or volume-averaged frameworks[37], which can couple ion transport and reaction kinetics but typically assume spatial uniformity and size-independent properties—assumptions that break down under nanoconfinement[38]. We hope these insights will inspire the development of future multiscale reaction-transport models—a task that remains both computationally demanding and methodologically underdeveloped.

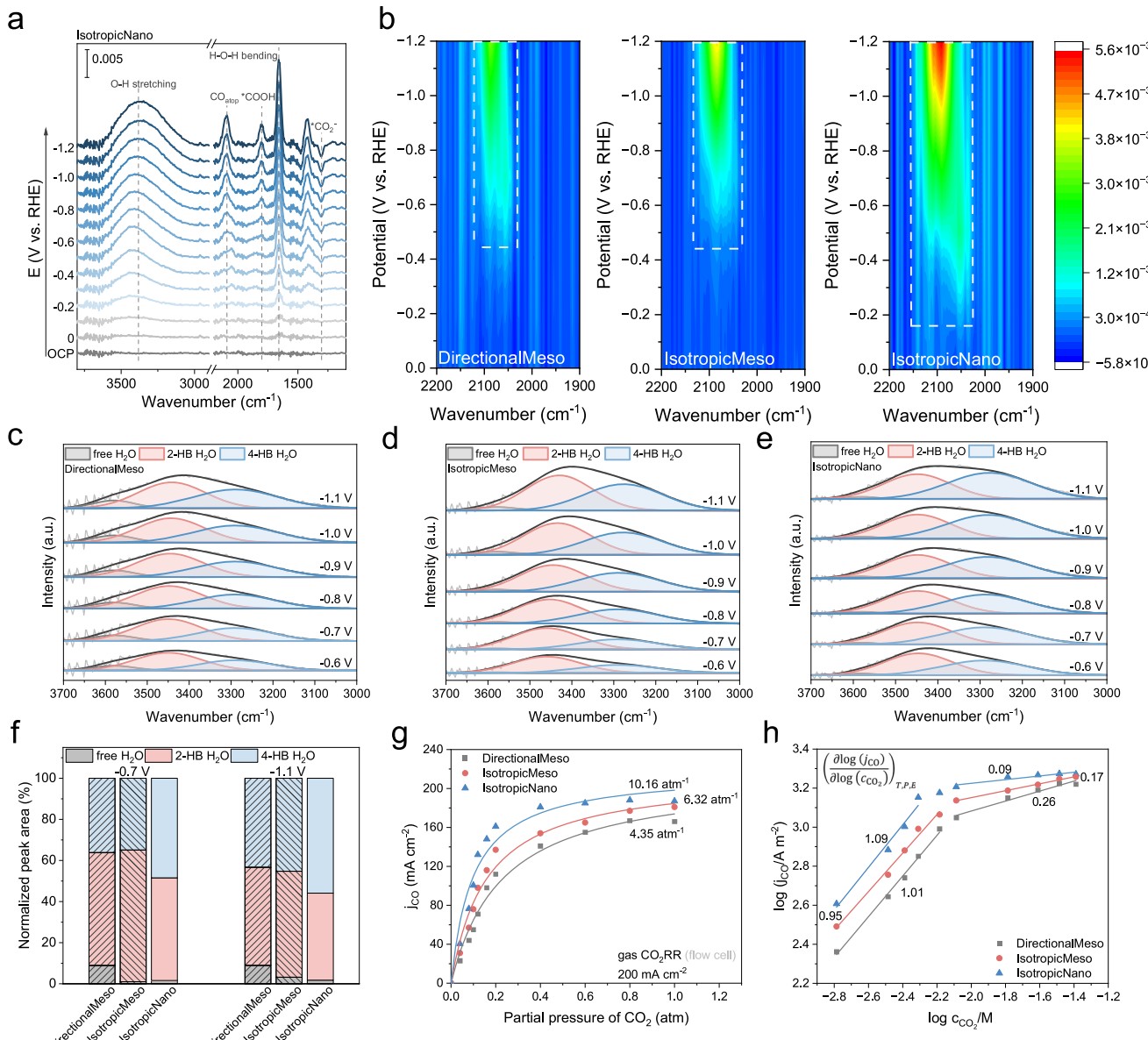

**Fig. 4 | Mechanistic study of confinement effects in nanopores. a** In situ ATR-SEIRS spectra recorded on IsotropicNano catalyst at OCP and under applied potentials ranging from 0 to −1.2 V versus RHE. 20% $CO_2$ gas (balanced by Ar) was purged into the system to form a $CO_2$-satured solution. **b** Contour map of in situ ATR-SEIRS spectra on three catalysts: DirectionalMeso, IsotropicMeso, and IsotropicNano, at the wavenumber range of 1900 to 2200 $cm^{-1}$. **c–e** Potential (versus RHE)-dependent fitted bands of interfacial water (3000–3600 $cm^{-1}$) on DirectionalMeso, IsotropicMeso, and IsotropicNano, respectively. **f** Variation in the proportion of interfacial water as a function of potential (versus RHE) for the three catalysts. **g** partial current density of CO ($j_{CO}$) on three catalysts as a function of the $CO_2$ partial pressure ($P_{CO2}$) at 200 mA/cm². The partial pressure of $CO_2$ was adjusted by mixing $CO_2$ with Ar gas. The binding affinity constants of the catalysts are described in the inset by fitting $j_{CO} – P_{CO2}$ plots. **h** Reaction order fitting of $\log(j_{CO})$ versus $\log(c_{CO2})$, showing two distinct regimes: diffusion-limited and kinetically controlled. Source data for the results are provided as a Source Data file.

## Nanoconfined environments for enhanced $CO_2$ binding and activation

To investigate how $i$-$CO_2$ transport within pores influences $CO_2$ reduction kinetics in a reactive capture system, we tested the porous catalysts in a simulated low-concentration $CO_2$ electrolysis setup using $K_2CO_3$ as the electrolyte. A Michaelis-Menten-based kinetic model[39,40] was used to describe the $CO_2$-to-CO pathway on Ni-SAC catalysts, where $CO_2$ binds to Ni sites forming an intermediate (Ni– $CO_2•−$) that releases CO (Supplementary Note 5 and Table S8).

We derived a binding constant (Fig. 4g), $K = \frac{k_a}{k_d + k_{cat}}$, to represent the catalyst's $CO_2$ binding affinity, where $k_a$, $k_d$, and $k_{cat}$ are the rate constants for adsorption, desorption, and catalysis, respectively. By varying the $CO_2$ concentration from 5% to 100% and performing electrolysis at 200 mA/cm², we fitted the K values: IsotropicNano (10.2) > IsotropicMeso (6.3) > DirectionalMeso (4.4), indicating favorable $CO_2$ binding on IsotropicNano.

Reaction order analysis revealed two regimes (Fig. 4h): a diffusion-limited region where all samples exhibited a reaction order -1, and a kinetically controlled region. IsotropicNano showed a narrower diffusion-limited range (0–12% $CO_2$) than the others (0–20%), indicating more efficient $CO_2$ utilization under lean conditions. In the kinetically controlled regime, DirectionalMeso and IsotropicMeso displayed higher reaction orders (0.2–0.3), suggesting continued dependence on $CO_2$ concentration due to mass transport limitations. Temperature-dependent studies further showed that IsotropicNano

had the lowest activation energy, indicating superior charge transfer properties (Supplementary Fig. 44).

To gain molecular-level information on $CO_2$-to-CO conversion, we conducted in situ attenuated total reflectance–surface-enhanced infrared absorption spectroscopy (ATR-SEIRAS) under simulated low-concentration $CO_2$ conditions. Controlled potential experiments revealed characteristic peaks for $CO_2$ reduction intermediates (*COOH, *CO) and interfacial water across all three samples (Fig. 4a and Supplementary Fig. 45).

The linearly adsorbed *CO ($CO_{atop}$) peak (2000–2100 $cm^{-1}$)[41,42] increased in intensity at more negative potentials (Fig. 4b). IsotropicNano showed ~200 mV lower overpotential for *CO appearance compared to DirectionalMeso and IsotropicMeso, indicating enhanced *COOH stabilization or earlier *CO formation. From −0.2 to −1.2 V (vs. RHE), IsotropicNano exhibited a blue shift in *CO stretching (2049 → 2100 $cm^{-1}$), suggesting weakened CO binding—likely due to reduced back-donation from metal d-orbitals to CO's π orbital, which favors *CO desorption and reduces surface poisoning[43]. A Stark tuning rate of ~51 $cm^{-1}$/V, observed on IsotropicNano, highlights its sensitivity to interfacial electric fields and its facilitation of *CO desorption[41]. In contrast, *CO peaks on DirectionalMeso and IsotropicMeso exhibited minimal Stark tuning or remained unchanged, indicating stronger binding and limited desorption capability.

We also monitored the O–H stretching region (~3000–3600 $cm^{-1}$) to probe hydrogen bonding networks and interfacial water structure (Fig. 4a and Supplementary Fig. 45). All samples showed a red shift of peaks at more negative potentials, indicating increased hydrogen bonding and enhanced water structuring[44]. Such structured H-bond networks can stabilize key intermediates (*COOH or $CO_2$) through polar interactions, lower the activation barrier for $CO_2$ reduction, and help sustain a local proton source without promoting excessive HER[45].

Deconvolution revealed peaks at 3600, 3450, and 3270 $cm^{-1}$, corresponding to free water, weakly hydrogen-bonded (2-HB), and strongly hydrogen-bonded (4-HB) species, respectively (Fig. 4c–f and Supplementary Table 9)[46,47]. Free water decreased significantly at high overpotentials on all samples. Notably, IsotropicNano showed a higher 4-HB/2-HB ratio and stronger spectral shifts, suggesting more robust H-bond networks that facilitate proton-coupled electron transfer (PCET) and intermediate stabilization during $CO_2$ reduction.

Taken together, the electrokinetic and in situ spectroscopic data indicate that IsotropicNano's nanoconfined 3D pore network enhances $CO_2$ binding affinity, stabilizes key intermediates, facilitates CO desorption, and promotes proton activation—collectively improving CO production. These tailored local reaction environments are resulted from the nanoconfined spaces that concentrated local reactants, which enhanced local electric field, tailored double layer structure, and the layout or binding angles of $CO_2$ and $H_2O$[35,48].

### Optimizing reactive capture system to increase energy efficiency

In the carbonate-liquid-fed reactive capture system, we employed IsotropicNano as the cathode catalyst in a BPM electrolyzer (Supplementary Fig. 46). A hydrophilic interposer layer was introduced between the CEL and catalyst layer to establish a pH gradient, ensuring a low pH at the CEL for efficient $i$-$CO_2$ generation and a higher pH at the catalyst layer to facilitate $CO_2$RR[14]. We optimized the interposer thickness, pore size, and material type, identifying the hydrophilic mixed cellulose ester (MCE) with a 135 μm thickness as achieving the highest $FE_{CO}$ (Supplementary Fig. 47).

In terms of DAC capture kinetics (using concentrated KOH) and subsequent reactive capture product selectivity, we selected 1.5 M $K_2CO_3$ as the feed solution (Supplementary Fig. 48). We observed that $FE_{CO}$ is maintained >40% at 100–400 mA/cm² (Fig. 5a), with a peak $FE_{CO}$ of 50% ± 3% achieved at 300 mA/cm². The $FE_{CO}$ thus appreciably

outperforms the best-prior $FE_{CO}$ of 20% at 300 mA/cm² in BPM-based carbonate electrolyzers[12].

The energy efficiency (EE) depends also on the full cell voltage. The baseline case, with its $FE_{CO}$ = 30% and $V_{cell}$ = 3.4 V at 100 mA/cm², corresponds to an energy cost of 64 GJ/tonne of 2:1 syngas. This projected energy consumption assumes the use of an 80%-efficient a solid oxide electrolyzer cell (SOEC) to balance CO to the final syngas composition of 2:1 $H_2$: CO.

We aimed therefore to pursue higher EE by lowering voltage. To study the origins of high voltage, we employed a diagnostic electrolyzer, focusing on the 300 mA/cm² case which required $V_{cell}$ = 5.1 V (Fig. 5c and Supplementary Fig. 49). The BPM accounted for 53% of the total $V_{cell}$ (Supplementary Fig. 49). When we replaced the BPM with one employing a nanoparticle $SnO_2$ WD catalyst[49] and a NiFeP anode, $V_{cell}$ declined to 3.3 V at 300 mA/cm². Uniting the best catalyst with the best BPM and using a 65%-efficient water electrolyzer to supply the missing $H_2$ to syngas composition of 2:1 $H_2$: CO, we obtained a syngas EE of 46% at 300 mA/cm² (Fig. 5c), surpassing the previous best EE of 28% at the same current density[12].

To examine the operating stability of the full electrolyzer with IsotropicNano cathode, $SnO_2$-BPM, and NiFeP anode, we characterized the system at 200 mA/cm² over the course of 50 h of electrolysis (Fig. 5b). The system retained FE > 40% and $V_{cell}$ of ~3.0 V, but only if we replaced the hydrophilic MCE interposer layer after 15 h.

X-ray absorption near-edge structure (XANES) spectra showed similar Ni K edges before and after electrolysis (Supplementary Fig. 50a). X-ray absorption fine structure spectrometry (EXAFS) showed the local coordination environment of Ni atoms, with a dominant peak at around 1.4 Å, assignable to the Ni-N coordination (Supplementary Fig. 50b), indicating a stable Ni single-atom structure. This Ni-N coordination anchors Ni atoms within the carbon-nitrogen framework, preventing the aggregation and migration of Ni under biased conditions. Anchoring Ni within the porous matrix minimized the formation of Ni particles that might, if formed, block the nanopore entrance.

The combination of $FE_{CO}$, $V_{cell}$, EE, and stability is comparable to, or exceeds, those reported in previous reactive capture studies utilizing various capture media or targeting different products (Supplementary Table 10).

It is possible (and undesired) for $i$-$CO_2$ to diffuse all the way through the catalyst layer and emerge as $CO_2$ in the tailgas stream. To keep track of the efficiency with which $CO_2$ is instead desirably converted or utilized, we define:

$$\text{Carbon utilization} = \left(1 - \frac{n_{CO2(g)}}{n_{CO2}^0}\right) \times 100\% \qquad (1)$$

where $n_{CO2}^0$ is the number of moles of $CO_2$ generated by the action of the BPM; and $n_{CO2(g)}$ is the number of moles of gas $CO_2$ detected at the outlet of the catholyte. We observed carbon utilization >99% at 300 mA/cm², with $CO_2$ concentration in the cathode outlet <1%. The outlet CO concentration reached ~50% also at 300 mA/cm². This combination of carbon utilization and CO concentration is the highest —looking across both reactive capture and gas-fed $CO_2$ reduction systems (acidic, neutral, and basic conditions) –reported to date (Supplementary Table 11).

We conducted an energy analysis of systems of interest (Fig. 5d and Supplementary Note 6): i) sequential capture-and-release followed by gas-fed electrochemical $CO_2$ upgrade; ii) sequential capture-and-release followed by reverse water-gas shift (RWGS) using hydrogen from an efficient water electrolyzer; iii) the integrated reactive capture approach explored in this study and prior studies, with $H_2$ is supplemented from a water electrolyzer to obtain $H_2$:CO of 2:1 syngas. The integrated reactive capture process in this work offers the lowest projected energy expense in GJ/tonne of syngas (43 GJ/tonne), its

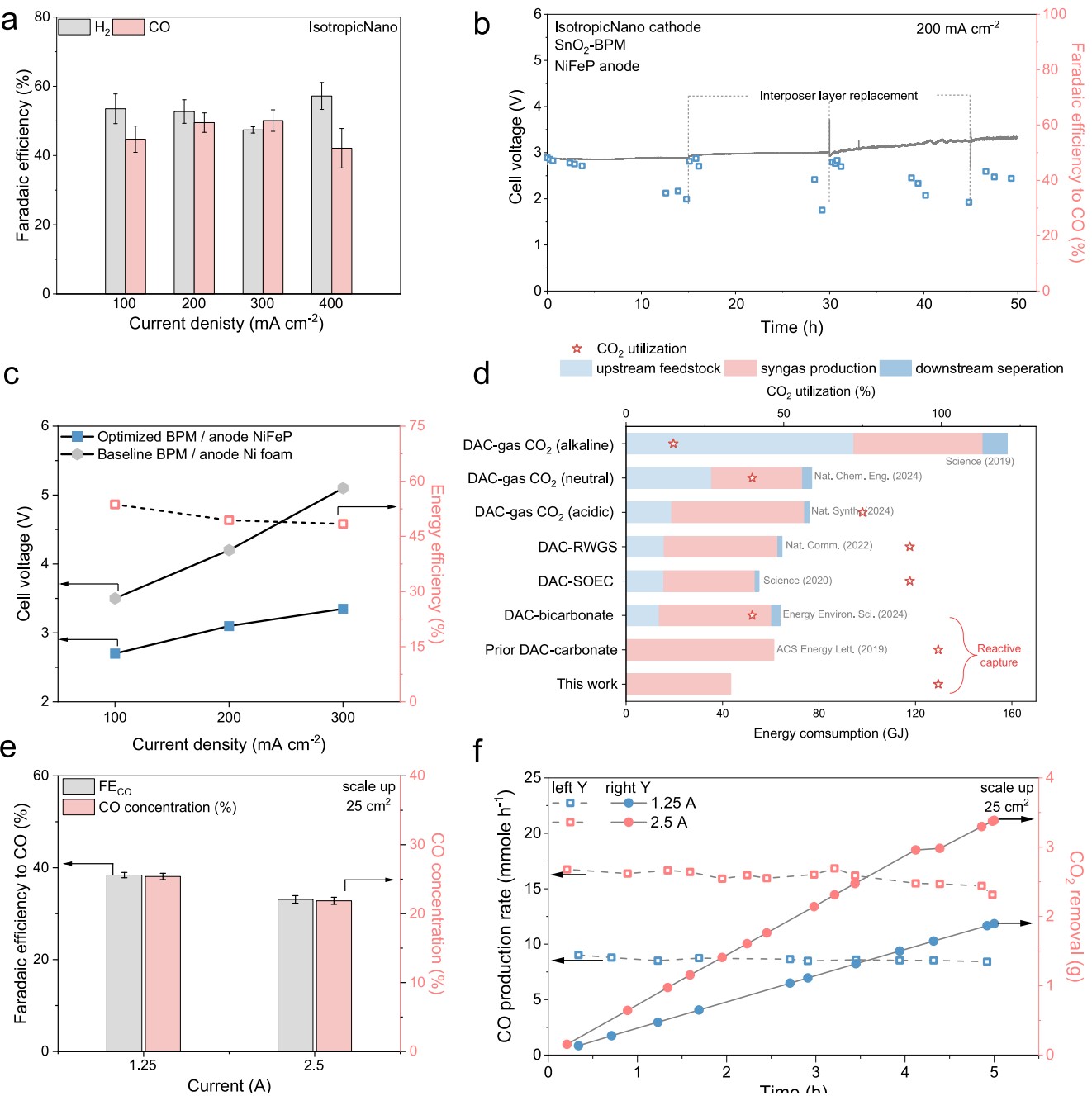

**Fig. 5 | Optimizing the reactive capture system for enhanced energy efficiency and scalability. a** Faradaic efficiency to CO and $H_2$ for IsotropicNano catalysts at 100–400 mA/cm². Where error bars are shown, values are means, and error bars indicate s.d. ($n = 3$ replicates). **b** Cell voltage (left Y-axis) and energy efficiency (right Y-axis) for carbonate electrolysis (1.5 M $K_2CO_3$) at different cell configurations. The energy efficiency is calculated from experimental data for a system that employed an optimized BPM and a NiFeP anode. **c** Cell voltage (left Y-axis) and Faradaic efficiency to CO (right Y-axis) for 50 h of electrolysis at 200 mA/cm² with an optimum cell configuration: IsotropicNano cathode, $SnO_2$-BPM, and NiFeP anode. **d** Comparison of energy consumption for the generation of 1 tonne syngas for different processes. In order, top to bottom, these are: direct air capture (DAC) coupled with gas $CO_2$ electrolysis in alkaline[59], neutral[60], and acidic[61] conditions; DAC-coupled with reverse water gas shift (RWGS)[62]; DAC coupled with a solid oxide electrolysis cell (SOEC)[63]; bicarbonate electrolysis best prior performance[21]; carbonate best prior performance[12]; and carbonate electrolysis reported in this work. **e** Faradaic efficiency to CO (left Y-axis) and CO concentration (right Y-axis) at the cathode outlet in the scaled electrolyzer. In the scaled flow plates, the channel depth is 1 mm, width is 1 mm, and the rid distance is 3.5 mm. The active area length is 50 mm, resulting in an electrode active area of 25 cm². Where error bars are shown, values are means, and error bars indicate s.d. ($n = 3$ replicates). **f** CO production rate (mmol h-1, left Y-axis) and $CO_2$ removal amount (right Y-axis) during 5 h of electrolysis at currents of 1.25 A and 2.5 A. Source data for the results are provided as a Source Data file.

advantage deriving the avoidance of the $CO_2$ regeneration and circulation steps in sequential DAC-plus-electrolysis.

We further identified key differences between carbonate- and previously reported bicarbonate-based reactive capture systems[21,50] in terms of electrolysis environment, energy consumption, and capital cost (Supplementary Note 7)[10]. Bicarbonate-liquid-fed electrolyzers typically show high $FE_{CO}$ due to greater $i$-$CO_2$ availability and a near-neutral buffered environment that suppresses $i$-$CO_2$ consumption through local acid-base reactions; however, they exhibit lower $i$-$CO_2$ utilization compared to carbonate systems (40–60% vs. >99%)[21,50].

While flue gas capture (4–15% $CO_2$) via $K_2CO_3$ (aq) absorbent can produce a bicarbonate-rich solution, it is kinetically unfavorable for DAC due to the low $CO_2$ concentration (~400 ppm)[10,11]. Converting post-captured carbonate to bicarbonate for continuous electrolysis adds extra processing steps, increasing both capital and operating costs[11]. Therefore, improving selectivity and lowering cell voltage in carbonate-based systems is still critical for efficient air-to-product CCU.

Finally, we carried out an initial scaling study, employing a 25 $cm^2$ BPM at 25 $cm^2$ catalyst-on-support (Supplementary Fig. 51). At currents of 1.25 A (50 mA/$cm^2$) and 2.5 A (100 mA/$cm^2$), $FE_{CO}$ was 38% and 33% (Fig. 5e). The $CO_2$ concentration in the cathode outlet remained <1% (Fig. 5e). We note that a low $CO_2$ concentration in the outlet tailgas will minimize costs of $CO_2$/CO separation.

Additionally, we recorded the time-averaged rate of CO production, observing 8.6 mmol h$^{-1}$ at 50 mA/$cm^2$ and 16 mmol h$^{-1}$ at 100 mA/$cm^2$, with each sustained over the course of 5 h of operation (Fig. 5f). This corresponds to the removal of 1.9 and 3.4 g of $CO_2$ from the carbonate capture liquid (Fig. 5f) from an input post-capture liquid that contained 13.2 grams of $CO_2$.

## Discussion

In this work, we employed a template synthesis strategy to develop nanoconfined Ni single-atom catalysts with isotropic, periodically aligned pore architectures, enabling efficient retention, binding, and activation of $i$-$CO_2$ in carbonate-fed reactive capture systems. The nanoconfinement effect, driven by short-range, non-electrostatic interactions between $CO_2$ and pore walls, enhances $i$-$CO_2$ accumulation and binding, intermediate stabilization, and *CO desorption, thereby boosting CO production. Through combined catalyst and system optimization, we achieved a high $FE_{CO}$ of 50% ± 3% at 300 mA/$cm^2$, >99% carbon utilization, and <1% $CO_2$ in the outlet, and a record EE of 46% for 2:1 syngas. These results highlight the role of interconnected nanopores and anchored single-atom sites in controlling $i$-$CO_2$ transport and reactivity under hydrophilic, reactant-starved electrolysis conditions.

## Methods

### Chemicals and materials

Nickel(II) acetylacetonate (95%), carbon tetrachloride (99.9%), ethylenediamine (99.5%), Pluronic P123 (average Mn-5800), oleylamine (OAm, >70%), octyltrimethylammonium bromide (OTAB, >98%), ammonium fluoride (NH$_4$F, >98%), potassium carbonate (ACS reagent, ≥99.0%), and potassium hydroxide (ACS reagent, ≥85%) were purchased from Sigma-Aldrich. Tetraethyl orthosilicate (TEOS, 98%) was purchased from Gelest. Heptane (HPLC grade), $n$-butanol (ACS grade), hydrofluoric acid (trace metal grade), hydrochloric acid (trace metal grade), and nitric acid (trace metal grade) were purchased from Fisher Scientific. The bipolar membrane (Fumatech FBM) and hydrophilic carbon paper (Freudenberg H23) were purchased from Fuel Cell Store. Ni foam (1.6 mm Thick) was purchased from MTI Corporation. All chemicals were used as received without further purification. Deionized (DI) water (18.2 MΩ cm, Barnstead™ E-Pure™) was used for all experiments in this work.

### Synthesis of templates

Synthesis of SBA-15. For the SBA-15 template synthesis[51], Pluronic P123 (4.0 g) was dissolved in 120 mL of 2.0 M HCl solution at 37 °C in a 500 mL polypropylene bottle. Once fully dissolved, TEOS (8.5 g) was added, and the solution was stirred for 20 h at 37 °C. The mixture was then transferred to an oven and kept at 100 °C for 24 h under static conditions. The resulting white precipitate was recovered by filtration, washed twice with DI water/methanol (1:1, v/v), and air-dried in a hood for 3 days. The dried solid was calcined at 550 °C for 5 h in air with a ramping rate of 2 °C/min. For the synthesis short channel SBA-15[52], in a

500 mL polypropylene bottle, Pluronic P123 (2.4 g), NH$_4$F (6.9 mg), and 1.94 M HCl (80 mL) were added. Once fully dissolved, a pre-mixed solution of TEOS (5.5 mL) and heptane (0.663 g) was added to the bottle, and the mixture was stirred for 4 min before being left under static conditions for 20 h. The mixture was then transferred to an oven and kept at 100 °C for 24 h under static conditions. The resulting white precipitate was recovered by filtration, washed twice with a 1:1 (v/v) mixture of DI water and methanol, and air-dried in a hood for 3 days. The dried solid was calcined at 550 °C for 5 h in air with a ramping rate of 2 °C/min.

Synthesis of MCM-48. For the MCM-48 template synthesis[53], in a 500 mL round-bottom flask, OTAB (0.678 g) and F-127 (2.538 g) were dissolved in a mixture of ethanol (43.0 g), water (129.8 g), and ammonia solution (12.24 g, 30–33% NH$_3$ in H$_2$O). The mixture was stirred vigorously at 850 rpm until fully dissolved. Subsequently, TEOS (1.8 g) was quickly added to the flask. Upon addition, the mixture was stirred continuously for ~1.5 min, during which it changed from colorless to light blue. The reaction mixture was then left undisturbed at room temperature for 24 h. The resulting solid was collected by centrifugation at 8000 rpm for 10 min, washed three times with ethanol (35 mL each), and dried under vacuum at room temperature overnight. The obtained powder was then calcined at 550 °C for 6 h under air at a ramping rate of 5 °C/min.

Synthesis of KIT-6. For the KIT-6 template synthesis[54], Pluronic P123 (4.0 g) was dissolved in a mixture solution of 144 mL of DI water and 7.9 mL of 35 wt. % hydrochloric acid. After which, 4.0 mL of $n$-butanol was added. This mixture was then stirred vigorously for 1 h at 35 °C. Following this, 8.6 mL of TEOS was added, and the solution was stirred continuously for another 24 h. The mixture was then subjected to hydrothermal synthesis at 100 °C for 24 h. Afterward, the resultant solid was filtered and washed with DI water. The obtained solid was then calcined at 550 °C for 6 h.

### Synthesis of Ni-NACs within silica templates

To a 50 mL round bottom flask, 1.8 g of ethylenediamine and 25–100 mg of nickel (II) acetylacetonate were added and mixed for 5 min. 4.0 g of carbon tetrachloride was added and stirred for 5 min. Subsequently, 0.8 g of silica template (SBA-15, KIT-6, or MCM-48) was introduced, and the mixture was refluxed at 90 °C for 16 h before being heated at 130 °C for 2 h to remove any residual volatiles. The resultant residue was then heated under a flowing Argon atmosphere with a ramping rate of 3 °C/min to 800 °C and remained at 800 °C for 2 h. The resulting black powder was dispersed in a solution of 5 wt.% HF and 10 wt.% HCl (35 mL) and stirred for 24 h. The remaining solid materials were collected by centrifugation, which was then washed with deionized water until a neutral pH was achieved. Finally, the final carbon material was dried at 80 °C overnight and stored for future use. The carbon materials synthesized using SBA-15, KIT-6, and MCM-48 are used directly as porous catalysts, named DirectionalMeso, IsotropicMeso, and IsotropicNano, respectively.

Post-synthesis thermal treatment of Ni-NAC was carried out under a diluted NH$_3$ gas atmosphere. Typically, the as-prepared Ni-NAC was annealed at 700 °C for 1 h under a flow of 10% NH$_3$ in He, with a heating rate of 5 °C/min. After annealing, the sample was naturally cooled to room temperature under the same gas atmosphere. The resulting material was used directly as the catalyst.

### Electrochemical measurements

The flow electrolyzer contains two stainless steel flow-field plates with serpentine channels, PTFE and silicone gaskets, and the MEA, which contains two electrodes and a membrane, and was formed after assembling the cell hardware. The catholyte and anolyte were circulated by peristaltic pumps (INTLLAB) at 20 ml min$^{-1}$. The applied current was controlled by a Bio-Logic VSP 300 potentiostat/galvanostat.

All reported cell voltages and electrode potentials are presented without iR compensation. The membrane used to separate catholyte and anolyte was a commercial (Fuel Cell Store) or custom-designed BPM. The as-prepared catalyst was used as the cathode. A piece of filter membrane was inserted as the interposer layer between cathode and cation exchange layer (CEL). The interposer layer was a hydrophilic mixed cellulose ester (MCE) membrane with a controlled thickness of 135 μm and a pore size of 8 μm. The catholyte was 1.5 M $K_2CO_3$, and the anolyte was 1 M KOH. All experiments were performed at room temperature.

The catalyst ink was prepared by dispersing Ni-NAC catalysts in 2-propanol solvent with added Nafion ionomer by ultrasonication. The ink needs to be well-sonicated for a good dispersion of catalyst. The mass ratio of the catalyst and ionomer was 9:1. The ink was then airbrushed onto the substrate to the final loading of ~1.5 mg cm$^{-2}$.

NiFeP electrode as anode was prepared from a modified method in the literature[55]. Ni foam was first cleaned by 6 M HCl and DI-water for 15 min under sonication. Then, a 40 mL solution with 4 mmol $NH_4F$, 10 mmol urea, 2 mmol $Ni(NO_3)_2 \cdot 6H_2O$, and 2 mmol $Fe(NO_3)_3 \cdot 9H_2O$ was prepared and transferred to a 50 mL Teflon-lined stainless steel autoclave. The hydrothermal growth of the hydroxides on Ni foam was performed at 120 °C for 6 h with a heating rate of 3 °C min$^{-1}$, followed by sonication in DI-water and drying in the oven at 80 °C to obtain NiFeOx catalyst. To prepare NiFeP, NiFeOx and 1.0 g of $NaH_2PO_2 \cdot H_2O$ were placed in a ceramic boat inside a tube furnace, with $NaH_2PO_2 \cdot H_2O$ positioned upstream relative to the gas flow. After purging with argon (Ar), the center of the furnace was elevated to 300 °C at a ramping rate of 1 °C min$^{-1}$ and kept at this temperature for 2 h in a static Ar atmosphere, and then naturally cooled down to ambient temperature.

The gas-phase $CO_2$ reduction reaction in a three-compartment flow cell was carried out using an Ag/AgCl (4 M KCl) reference electrode and a Ni foam counter electrode. The cathode catalysts were applied onto hydrophobic carbon paper (Freudenberg H23C3) via airbrushing. A 40-micron anion-exchange membrane (PiperION) was used to separate the cathode and anode chambers. Both the catholyte and anolyte consisted of 1.5 M $K_2CO_3$ solution.

For experiments involving varying $CO_2$ flow rates, a mass flow controller (Alicat Scientific) was used to regulate the gas flow. In studies with different $CO_2$ concentrations, $N_2$ was used as a balance gas to adjust the partial pressure of $CO_2$. The gases were mixed using a tee-type connector before being introduced into the flow cell. The total gas flow rates were measured with a gas flow meter. Product concentrations at the catholyte outlet were quantified using gas chromatography (GC).

For gas-phase $CO_2$ reduction in a membrane electrode assembly (MEA) setup, the device was constructed by sequentially assembling a cathode gas diffusion electrode (GDE, 4 cm² geometric area), a PiperION anion-exchange membrane, and an anode (IrO₂-GDE, 4 cm², Dioxide Materials). These components were secured within PTFE gaskets, each with a 4 cm² window, and the assembly was compressed between electrolyzer plates to ensure proper sealing, electron conductivity, and ionic transport.

Humidified $CO_2$ gas, controlled by a mass flow controller (Alicat Scientific), was introduced into the cathode flow field for the reaction. The product stream was mixed with a 2.5 mL/min $N_2$ stream to facilitate analysis by GC. On the anode side, a 0.1 M KHCO₃ solution was circulated at a flow rate of 20 mL/min. In MEA configurations operating at high current densities, significant $CO_2$ loss due to (bi)carbonate formation was mitigated by using the mixed $N_2$ stream as an internal standard for accuracy. We define the single-pass (SP) efficiency as follows:

$$\text{Single Pass Efficiency(SP)} : \text{SP} = \frac{\dot{V}_{CO}}{\dot{V}_{CO2,\,in}} \times 100\% \quad (2)$$

Where $\dot{V}_{CO}$ and $\dot{V}_{CO2,\,in}$ represent the volume flow rates of the produced CO and the total $CO_2$ inlet, respectively.

## Product analysis

The gas products ($H_2$ and CO) were quantified by GC (Shimadzu2014, PerkinElmer Clarus 580) equipped with a thermal conductivity detector (TCD) and a flame ionization detector (FID) equipped with a Methanizer. The calibration curve was established by analyzing the standard calibration gases with different concentrations (10–10,000 ppm). Argon (50 mL min$^{-1}$) was purged as the carrier gas to carry the gas products out of the system for quantification.

The rate of $H_2$/CO generation ($r$, mol s$^{-1}$) for each cycle was calculated by the following equation:

$$r = c \times 10^{-6} \times [P\dot{v} \times 10^{-6}/(RT)] \quad (3)$$

Where $c$ is the $H_2$/CO concentration (ppm); $\dot{V}$ is the volumetric flow rate of the inlet gas (100 ml min$^{-1}$); $p$ is the ambient pressure ($p = 1.013 \times 10^5$ Pa); $R$ is the gas constant ($R = 8.314$ J mol$^{-1}$ K$^{-1}$); $T$ is the room temperature (293.15 K). The total amount of gas (mol) was calculated by integrating the plot of $H_2$/CO production rate (mol s$^{-1}$) vs. reaction time (s).

Faradaic efficiency ($FE_i$) can be calculated by equations as follows:

$$FE_i = \frac{n_i z_i F}{Q} \times 100\% \quad (4)$$

Where $n_0$ is initial moles of reactant; $n$ is the moles of reactant after electrolysis; $n_i$ is the moles of product $i$; $z_i$ is the number of electrons transferred for one product molecule; $F$ is the Faraday constant (96,485 C mol$^{-1}$); $Q$ is the total charge passed through the electrolytic cell; t is the electrolysis time (s).

## Materials characterization

Field-Emission Scanning Electron Microscope (FE-SEM) was recorded on the FEI Teneo LoVac. Aberration-corrected HAADF STEM imaging was performed using a probe-corrected Thermo Fisher (FEI) Titan Themis. Powder X-ray diffraction (XRD) patterns of samples were recorded on a Bruker X-ray diffractometer using Cu Kα radiation (40 kV, 40 mA) over the range of 5–80 of 2θ°. X-ray photoelectron spectrometry (XPS) was recorded on a PerkinElmer PHI ESCA system with Physical Electronics (PHI). Nitrogen sorption isotherms were measured by the Micromeritics 3Flex analyzer at −196 °C. Before recording the $N_2$ sorption isotherms, the samples were pretreated at 200 °C under vacuum for 12 h. The total surface area was calculated using the Brunauer–Emmett–Teller (BET) equation. The micropore and mesopore surface area, and micropore volume were evaluated using the t-plot method. The nanopore size was determined using the Horvath-Kawazoe method, with measurements conducted on a Micromeritics Tristar instrument. The mesopore size was determined using the Horvath-Kawazoe method, with measurements taken on a Micromeritics 3Flex instrument. Inductively coupled plasma optical emission spectroscopy (ICP-OES) for nickel loadings was performed using an Agilent 5800 spectrometer. Elemental analysis of carbon and nitrogen was measured on a Thermo FlashSmart CHNS/O Elemental Analyzer. SAXS measurements were collected at beamline 12-ID-B of the Advanced Photon Source (APS) at Argonne National Laboratory. In situ ATR-SEIRAS spectroscopy was performed using a Thermo Scientific Nicolet iS50 spectrometer equipped with a custom-designed ATR electrochemical cell. The spectral resolution was set to better than 2 cm$^{-1}$, with 64 scans accumulated per spectrum. All other experimental parameters were configured according to the standard test protocol provided by the manufacturer or experimental requirements. The electrolysis was performed in a three-electrode system using an Ag/AgCl reference electrode, with $CO_2$-saturated 0.5 M KHCO₃ as the

electrolyte. The conversion of potentials from Ag/AgCl to the RHE scale was performed using the following equation:

$$E_{RHE} = E_{Ag/AgCl} + 0.197 + 0.059 \times pH \tag{5}$$

Electrochemical mass spectrometry (EC-MS) was conducted using our carbonate-fed gas-tight electrochemical cell coupled to a quadrupole mass spectrometer (Thermo Scientific™). The evolved gases were continuously directed from the headspace of the cathode chamber into the mass spectrometer via a heated capillary inlet maintained at 120 °C to prevent condensation. Signals corresponding to $CO_2$ (m/z = 44) were recorded under steady-state conditions at applied current densities ranging from 50 to 300 mA/cm². The background signal was subtracted.

## Computational methods

**The calculation of $CO_2$ adsorption energy to the Ni-SAC surface.** All density functional theory (DFT) calculations were performed using the Vienna ab initio simulation package (VASP). The Perdew-Burke-Ernzerh (PBE) exchange-correlation functional and the projector augmented wave (PAW) methods were adopted. The spin polarization was turned on for all simulations. The dispersion interactions were included using Grimme's D3 correction. The kinetic cutoff energy for the plane-wave basis set was set as 520 eV. We used a $k$-point sampling of $5 \times 5 \times 1$ with the Monkhorst-pack scheme for integration over the Brillouin zone in reciprocal space. The convergence criteria for the electronic and ionic loops were set at $10^{-5}$ eV and 0.02 eV Å$^{-1}$, respectively.

A set of supercells was constructed with a 20 Å-thick vacuum slab to separate the graphene layer from its periodic images in the $z$-direction. The graphene unit cell dimensions were optimized to be $a = b = 2.46$ Å and $\gamma = 120°$. The $4 \times 4$ supercell of graphene was used to construct the 4N-Gr support. A C-divacancy was created first. Then, four C atoms on the vacancy edge were replaced by four N atoms. After that, one Ni atom was embedded into the central coordination site to generate the Ni@4N-Gr structure. After the structural relaxation in DFT calculations, the Ni metal atom remains in the same plane as the C atoms. This configuration is consistent with the reported results[36,56].

The adsorption energy $E_{ads}$ of an adsorbed species (i.e., $CO_2$ or $H_2O$) on the Ni@4N-Gr is calculated as:

$$E_{ads} = E_{mol + Ni-SAC} - (E_{mol} + E_{Ni-SAC}) \tag{6}$$

where $E_{mol + Ni-SAC}$, $E_{Ni-SAC}$, and $E_{mol}$ are the energies of the catalysis with the molecule adsorbed, of the bare catalyst, and of the molecule, respectively.

**The calculation of $CO_2$ in-pore concentration and local number distribution in nanopores.** Figure 3b shows the representative modeling systems in our MD simulations. where a nanoslit formed by two parallel catalyst sheets was assumed to be rigid and in contact with aqueous electrolyte solution reservoirs at two sides. The slit walls have a dimension of $3.34 \times 2.99$ nm laterally, and the width of the reservoir is around 12 nm. Periodic boundary conditions were applied in all three directions. Different slit sizes were studied (defined as the distance between the atomic centres of neighbored graphene sheets), including 10.0 nm, 1.5 nm, and 1.0 nm, respectively. The aqueous electrolyte solution was composed of $CO_2$ and water. The ionic charge values were determined using the DDEC6 analysis[57] based on the DFT calculation results (summarized in Supplementary Table 6). To model the $CO_2$ adsorption affinity towards catalyst surfaces, we optimized the force field parameters using an in-house generic algorithm code (GA)[58]. The specific parameter values are summarized in the Supplementary Table 7.

## Data availability

All the data that support the findings of this study are available in the main text and the Supplementary Information, or from the corresponding authors upon reasonable request. Source Data file for DFT and MD structures has been deposited in Figshare under accession code DOI link https://doi.org/10.6084/m9.figshare.29323862.v1. Source data are provided with this paper.

## Code availability

All data and information for the simulations are available in the supplementary materials. Any additional information is available from the corresponding authors upon a reasonable request.

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

## Acknowledgements

This work is supported by Office of Naval Research under the award number of N00014-22-1-2690. L.A., J.Z., W.H., and L.Q. were supported by the Laboratory-Directed Research and Development program at Ames National Laboratory for the design and synthesis of carbon materials. J.Z. and L.Q. were supported by the U.S. Department of Energy (DOE), Office of Basic Energy Sciences, Division of Chemical Sciences, Geosciences, and Biosciences, Catalysis Science program, for the characterization of carbon materials and data analysis. The Ames Laboratory is operated for the U.S. DOE by Iowa State University under contract no. DE-AC02-07CH11358. This work is also supported by the research grant from ARC DP210103888 and FT220100149. This research used resources of the Advanced Photon Source (beamline 12-ID-B), a U.S. DOE Office of Science User Facility, operated for the DOE Office of Science by Argonne National Laboratory under Contract No. DE-AC02-06CH11357

## Author contributions

E.H.S., K.X., L.Q., and J.Z.L. supervised the project. H.L. conceived the key ideas and analysis. H.L. and C.Y. carried out the electrochemical experiments and data analysis. L.A. and J.Z. synthesized the catalysts. Peiyao W. performed the computational analysis. H.L., L.A., C.Y., J.Z., H.S., B.P., J.L., M.L., H.A., J.Y., Y.C., Peiying W., K.-S.L., K.L., and Z.L. contributed to catalyst characterization. O.K.F., W.H., J.Z.L., L.Q., K.X., and E.H.S. provided input on manuscript editing and discussions. All authors contributed to data interpretation and manuscript revision.

## Competing interests

The authors declare no competing interests.
