## [Transparent Peer Review file · Nature Communications]

Interconnected Nanoconfining Pore Networks Enhance Catalyst CO₂ Interaction in Electrified Reactive Capture

Corresponding Author: Professor Edward Sargent

Version 0:

Reviewer comments:

Reviewer #1

(Remarks to the Author)

This work reports a design of CO₂ generation from the reaction of carbonate and H⁺ provided via the cation exchange side of a bipolar membrane and electrochemical reduction of in situ-generated CO₂. In fact, a similar idea for the CO₂ capture into (bi)carbonate and converting (bi)carbonate into syngas or even format as well as the related economic analysis has been systemically studied in many works from Berlinguette group (ACS Energy Lett. 2020, 5, 7, 2165–2173. ACS Energy Lett. 2020, 5, 8, 2624–2630. Applied Energy 325, 119897). In 2016, Berlinguette has already demonstrated the electrolysis of CO₂ to syngas in a bipolar membrane-based cell (DOI: 10.1021/acseenergylett.6b00475). Thus, the idea of such a BPM system for the reduction of CO₂ is not novel. However, this manuscript demonstrates that a nanopore cathode enables the accumulation of in situ-generated CO₂ inside the pore structure, which enhances the interaction of CO₂ with the active site, promoting CO selectivity at high current densities. This cathode design is quite interesting, and the authors also made the design of this BPM system with 25 cm². However, I feel this manuscript is not well-written, thus it is not so readable. The conclusion part of this work is even missing. After major revisions and significant polishing of the text and structure, this work may be reconsidered.

More comments:

It is not clear why the authors used 1.5 M carbonate electrolytes. It should be interesting to show the carbonate concentration effect on the catalytic selectivity at identical current density (maybe 200 Ma/cm²).

I do not think it is appropriate to call 'carbonate electrolysis', because carbonate is not reduced electrochemically, and it is not involved in any electrolysis directly here. It is still CO₂ electrolysis, thereby so-called 'Carbonate electrolysis' may mislead the readers.

Line 194, it is not clear what 'single-pass efficiency' means.

The function of the interposer (filter membrane) in the electrolyzer is not clear, and I think it is better to have a clear explanation and description of the importance of the interposer.

I am also wondering if the thickness of the filter membrane (interposer) could correlate with CO₂ generation and transport, which affects the catalytic selectivity.

Reviewer #2

(Remarks to the Author)

The manuscript titled "A Permeable Porous Conductive Catalyst Increases the Catalyst:CO₂ Interaction Cross-Section in Electrified Reactive Capture" presents a study on a Ni single-atom catalyst dispersed in a conductive carbon support, forming a permeable porous conductive catalyst (P2C2) intended to enhance CO₂ retention and electroreduction efficiency in an electrified reactive capture system using carbonate capture liquid. While the topic is scientifically relevant and aligns with the scope of the journal, the manuscript suffers from several critical issues that significantly undermine its novelty and scientific contribution. Therefore, I recommend rejecting this submission. The major concerns are outlined as follows:

1. Lack of Novelty: The concept of employing a porous medium catalyst in a carbonate capture liquid electrochemical CO₂

reduction system has already been reported by Lee et al. (Geonhui Lee, Armin Sedighian Rasouli, Byoung-Hoon Lee, Jinqiang Zhang, Da Hye Won, Yurou Celine Xiao, Jonathan P. Edwards, et al., *Joule*, 2023, 7(6): 1277-1288). The primary distinction claimed in this manuscript is the introduction of a Ni single-atom catalyst. However, the authors fail to demonstrate the unique advantages of this catalyst over previously reported systems. There is insufficient discussion and quantitative comparison regarding performance improvements, such as selectivity enhancement, reaction efficiency, or suppression of side products.

2. Inadequate Modeling of CO₂ Transport: The section titled “Simulation of i-CO₂ transport kinetics in pores” only considers the physical adsorption and diffusion of CO₂ within the pores, without coupling the electrochemical reduction reaction. The dynamic interaction between CO₂ mass transport and catalytic reaction is a critical aspect in such confined pore systems, and neglecting this coupling renders the simulation results incomplete and of limited practical value. Furthermore, the impact of porous structure on CO₂ transport and adsorption behavior has been extensively studied in prior works, diminishing the originality of this specific modeling approach.

3. Superficial Data Analysis: Although the authors compare the performance of different porous catalysts, the interpretation of structure-performance relationships is underdeveloped. There is a lack of in-depth analysis of the potential synergistic effects between the microporous structure and the Ni single-atom active sites. For instance, it remains unclear whether the micropores merely serve as CO₂ retention zones or actively modify the electronic structure of the Ni sites. Additionally, the explanation for the observed improvement in Faradaic efficiency is largely qualitative, with limited kinetic data or in situ characterization to substantiate the claims.

4. Insufficient Literature Contextualization: The manuscript provides an incomplete review of the state-of-the-art research in carbonate capture liquid-based CO₂ electroreduction. Recent developments involving Cu-based catalysts and other transition metal single-atom catalysts for selective C₂₊ production in similar systems have not been adequately discussed. This oversight weakens the manuscript’s positioning and reduces its clarity in highlighting the research gap it aims to address.

Overall, the manuscript, while addressing a relevant problem, lacks the necessary scientific novelty, rigor in coupling experimental and theoretical analyses, and a comprehensive positioning within the existing literature. Given these substantial limitations, I recommend rejecting the manuscript in its current form.

Reviewer #3

(Remarks to the Author)

Liu et al. developed several porous carbon materials as supports for nickel single-atom catalysts in a carbonate-fed reactive capture system. The authors also employed the state-of-the-art bipolar membrane and anode material, the system achieved good performance. The design and fabrication of the porous carbon materials appear rational, with improved porosity, the interaction between the catalyst and in-situ CO₂ molecules could be enhanced. I have a few questions for the authors:

1. The authors mentioned that CO₂ retention is a more important factor compared to nickel content in these catalysts. Did the authors try higher nickel loadings on these carbon materials, and how were the results?
2. Nickel single atoms have been reported to be selective for CO₂-to-CO conversion in acidic conditions. Did the authors test the cell without the interposer?
3. Some studies suggest that nitrogen doping contributes to CO₂ reduction by providing active sites for CO₂ adsorption and conversion, with different nitrogen species exhibiting different activities. The authors used four different nitrogen sources, did this contribute to the differing performances of the final samples?

Reviewer #4

(Remarks to the Author)

In this manuscript, the author presents a catalyst/electrode design that exhibits enhanced performance in the electroreduction of a carbonate solution into a CO+H₂ mixture. Although the performance improvement is notable compared to previous studies, the reviewer finds that the work lacks sufficient novelty, fundamental understanding, and engineering advancement. Consequently, it is not deemed suitable for publication in *Nature Communications*. It may be appropriate for journals such as *EES Catalysis* or *Chem Catalysis*.

The concept of a “Permeable Porous Conductive Catalyst” is overhyped and actually lacks novelty. Here, “permeable” implies that the gas produced in the liquid phase can penetrate the electrode and be separately collected. However, this phenomenon is common to nearly all catalysts deposited on carbon paper—whether they are nano-sized materials, micro-sized particles, or even sputtered films. Moreover, many studies have already deposited porous, conductive materials—using methods such as templating or dealloying—onto carbon paper and evaluated their performance in CO₂/CO electroreduction. Although these studies focus on the electrolysis of CO₂/CO instead of carbonate, their existence renders the catalyst/electrode design concept presented in this work insufficiently novel.

A substantial portion of this high-surface-area, porous electrode may not contribute to performance in carbonate electrolysis.

The authors suggest that the porous catalyst structure aids in retaining and accumulating CO₂, formed via carbonate acidification, within its pores, thereby enhancing the reduction rate. However, under actual reaction conditions, the by-product OH⁻, generated during CO₂ electroreduction to CO, accumulates significantly within these pores and elevates the local pH. As a result, incoming CO₂ reacts readily with OH⁻—a kinetically more favorable process than the electroreduction on the catalyst surface. This effect is notable even in planar electrodes at the current densities used in the study and is markedly intensified within nanopores. Consequently, only the top layer of the electrode can actively participate in CO₂ reduction, leaving the majority of the active sites inaccessible—a well-documented transport limitation in many electrochemical reactions occurring in porous electrodes.

In fact, many previous studies on CO₂ electroreduction to formate represent a significantly more efficient approach for converting carbonate into value-added products. Whether using an H-type cell or a microfluidic flow cell with catalysts such as Sn, Pb, In, or Bi-based materials, as long as a bicarbonate solution—readily obtained by saturating a carbonate solution with CO₂—is employed as the electrolyte, the process essentially converts bicarbonate into formate. This method achieves much higher Faradaic efficiencies and current densities. Previous researchers simply did not market these studies as efficient conversions of alkaline-captured CO₂. Although syngas may be more widely demanded than formate, from a scientific standpoint, the concept in the current manuscript does not contain sufficient novelty.

Version 1:

Reviewer comments:

Reviewer #1

(Remarks to the Author)

The authors have addressed my concerns properly, and I have no more comments.

Reviewer #2

(Remarks to the Author)

All the concerns raised have been adequately addressed.

Reviewer #3

(Remarks to the Author)

Thanks for the responses. The result of nickel loading study is helpful. The authors mention increasing Ni content may promote nanoparticles formation, did the authors observe this through TEM imaging?

Reviewer #4

(Remarks to the Author)

The authors provided an unnecessarily lengthy response yet failed to directly address this reviewer's straightforward comments, ultimately failing to persuade.

In their response to comment 1, the authors acknowledged that their electrode structure is already common in CO₂ electrochemistry. They then shifted their argument to highlight the conductive porous networks of the catalyst. However, they did not provide convincing evidence that these convoluted nanopores actually enhance the reduction reaction. Particularly in their response to comment 2, the authors admitted that approximately 50-80% of the CO₂ reacts with OH⁻ generated during surface reduction, strongly suggesting these pores are inaccessible to CO₂ from the electrolyte—a well-known limitation in electrolysis involving gaseous reactants with restricted mass transfer.

In response to comment 3, when comparing their system to a CO₂-bicarbonate system, the authors inaccurately stated, "These systems require energy-intensive CO₂ regeneration, circulation, and downstream separation after CO₂ capture." This claim is fundamentally incorrect because: (1) CO₂ regeneration is not required in such systems, and CO₂ is not consumed significantly, thus eliminating substantial sourcing costs; and (2) circulating CO₂ at atmospheric pressure is not energy-intensive.

In summary, the authors have overstated concepts that are neither fundamentally novel nor practically impactful, and their results remain mediocre.

Version 2:

Reviewer comments:

Reviewer #3

(Remarks to the Author)

The authors have addressed all my questions and comments.

Reviewer #4

(Remarks to the Author)

This reviewer will not further comment and reserves judgment on the scientific and practical contributions of this work.

Response to Reviewer's Comments

Prior Title: A Permeable Porous Conductive Catalyst Increases the Catalyst:CO₂ Interaction Cross-Section in Electrified Reactive Capture

Current Title: Interconnected Nanoconfining Pore Networks Enhance Catalyst:CO₂ Interaction in Electrified Reactive Capture

Manuscript ID: NCOMMS-25-07597-T

We thank all reviewers for their deep engagement with this work and helpful inputs. We have acted on each comment and significantly revised the work.

A point-by-point response to the comments is provided below. In the revised manuscript, we use yellow text to indicate additions and changes made in light of referee advice.

Reviewers' Comments:

Reviewer #1 (Remarks to the Author)

This work reports a design of CO₂ generation from the reaction of carbonate and H⁺ provided via the cation exchange side of a bipolar membrane and electrochemical reduction of in situ-generated CO₂. In fact, a similar idea for the CO₂ capture into (bi)carbonate and converting (bi)carbonate into syngas or even format as well as the related economic analysis has been systemically studied in many works from Berlinguette group (ACS Energy Lett. 2020, 5, 7, 2165–2173. ACS Energy Lett. 2020, 5, 8, 2624–2630. Applied Energy 325, 119897). In 2016, Berlinguette has already demonstrated the electrolysis of CO₂ to syngas in a bipolar membrane-based cell (DOI: 10.1021/acsenergylett.6b00475). Thus, the idea of such a BPM system for the reduction of CO₂ is not novel. However, this manuscript demonstrates that a nanopore cathode enables the accumulation of in situ-generated CO₂ inside the pore structure, which enhances the interaction of CO₂ with the active site, promoting CO selectivity at high current densities. This cathode design is quite interesting, and the authors also made the design of this BPM system with 25 cm². However, I feel this manuscript is not well-written, thus it is not so readable. The conclusion part of this work is even missing. After major revisions and significant polishing of the text and structure, this work may be reconsidered.

Response: We appreciate the reviewer's valuable feedback on the BPM-based system for reactive capture, and point out the interest of our cathode design.

While previous studies by the Berlinguette group have indeed explored BPM-based systems and bicarbonate electrolysis, our study focuses on **carbonate-fed** electrolysis systems, which present distinct thermodynamic, kinetic, and electrochemical challenges compared to bicarbonate systems. We emphasize below the key differences:

1. CO₂ capture scenario and thermodynamics

- KOH-based **direct air capture (DAC)**: Requires concentrated KOH as capture liquid due to low CO₂ levels (~400 ppm), producing a K₂CO₃-rich post capture solution ([10.1016/j.joule.2018.05.006](https://doi.org/10.1016/j.joule.2018.05.006)).
- K₂CO₃-based **flue gas capture**: Can use K₂CO₃ as the absorbent since CO₂ concentrations are higher (4–15%), resulting in a KHCO₃-rich post capture solution.

2. Kinetics and practical considerations

- Capturing CO₂ with KOH to form K₂CO₃ (pH ~12) is feasible for DAC, but further conversion to KHCO₃ (pH ~8) is limited by slower kinetics at low CO₂ levels (~400 ppm) and requires up to 14× more contactors, making KHCO₃-based approaches impractical without major cost reductions (<https://pubs.acs.org/doi/10.1021/acsenergylett.4c00807>).
- Transitioning from K₂CO₃ to KHCO₃ feedstocks for electrolysis, e.g., via pH downshifter BPMED, adds complexity and increases both energy and capital costs.

(3) Electrochemical environment and *i*-CO₂ utilization

Reactive capture system	H ⁺ needed for one i -CO ₂	i -CO ₂ utilization (%)	bulk pH	local pH during electrolysis at the catalyst layer
Bicarbonate	1	40-60	8.5	9-11
Carbonate	2	>99	~12	13-14

- Carbonate-fed systems require twice the amount of H⁺ per mole of *i*-CO₂ generated, presenting more stringent challenges in *i*-CO₂ generation and supply.
- Bicarbonate electrolyzers, while exhibiting higher *i*-CO₂ availability and favorable buffering that lower and stabilize local pH, suffer from spontaneous decomposition and *i*-CO₂ loss, leading to lower *i*-CO₂ utilization.
- In contrast, carbonate-fed systems enable nearly complete carbon utilization with minimal *i*-CO₂ loss in the outlet stream, simplifying downstream gas separation and reducing total process costs.

These differences are now more explicitly discussed in **Supplementary Note 7**.

On page 3 of the manuscript, we now add:

“Considering the low CO₂ concentration in air (~400 ppm), the post-capture liquid from DAC using KOH as the liquid absorbent, if no further processing or treatment is applied, is primarily a carbonate-rich solution.^{5,11”}

On page 16 of the manuscript, we now add:

*“We further identified key differences between carbonate- and previously reported bicarbonate-based reactive capture systems^{21,44} in terms of electrolysis environment, energy consumption, and capital cost (**Supplementary Note 7**).¹⁰ Bicarbonate-liquid-fed electrolyzers typically show high FE_{CO} due to greater i-CO₂ availability and a near-neutral buffered environment that suppresses i-CO₂ consumption through local acid-base reactions ; however, they exhibit lower i-CO₂ utilization compared to carbonate systems (40–60% vs. >99%).^{21,44} While flue gas capture (4–15% CO₂) via K₂CO₃ (aq) absorbent can produce a bicarbonate-rich solution, it is kinetically unfavorable for direct air capture (DAC) due to the low CO₂ concentration (~400 ppm).^{10,11} Converting post-captured carbonate to bicarbonate for continuous electrolysis adds extra processing steps, increasing both capital and operating costs.¹¹ Therefore, improving selectivity and lowering cell voltage in carbonate-based systems is still critical for efficient air-to-product CCU.”*

Novelty of our approach

The key contribution of our work lies in the nanoconfined cathode design, which enables i-CO₂ accumulation and selective CO formation under the reactant-starved, alkaline conditions of carbonate-fed BPM systems. We show that the isotropic, ordered nanoporous architecture enhances CO₂ transport, intermediate stabilization, and *CO desorption, leading to >99% carbon utilization and high energy efficiency without the need for gas separation.

Additionally, we have revised the manuscript to enhance its readability. The **conclusion** section has been added to **page 17** in the revised MS.

“Conclusion

*In this work, we employed a template synthesis strategy to develop nanoconfined Ni single-atom catalysts with isotropic, periodically aligned pore architectures, enabling efficient retention, binding, and activation of i-CO₂ in carbonate-fed reactive capture systems. The nanoconfinement effect, driven by short-range, non-electrostatic interactions between CO₂ and pore walls, enhances i-CO₂ accumulation and binding, intermediate stabilization, and *CO desorption, thereby boosting CO production. Through combined catalyst and system optimization, we achieved a high FE_{CO} of 50% ± 3% at 300 mA/cm², >99% carbon utilization and <1% CO₂ in the outlet, and a record*

energy efficiency of 46% for 2:1 syngas. These results highlight the critical role of nanopores and anchored single-atom sites in controlling *i*-CO₂ transport and reactivity under hydrophilic, reactant-starved electrolysis conditions.”

Manuscript Structure and Writing Flow:

To enhance readability while maintaining concise language, the manuscript follows a logical progression in the following subsections:

- **Nickel single-atom catalysts with engineered pore structure:** Catalyst synthesis and characterization to elucidate pore architectures and their structural properties
- **Electrolysis performance as a function of pore structure:** compare the electrochemical performance in carbonate-fed electrolysis, compare the influence of pore structure on catalyst layer utilization and *i*-CO₂ utilizations
- **Characterization of *i*-CO₂ transport kinetics in pores:** experimental investigations on *i*-CO₂ transport behaviors, including *i*-CO₂ adsorption, retention, and active site accessibility in CO₂-starved environments
- **Simulation of *i*-CO₂ transport kinetics in pores:** DFT and MD simulations on molecular-level insights into *i*-CO₂ diffusion and interactions within nanoconfined pores
- **Nanoconfined environments for enhanced CO₂ binding and activation:** In situ spectroscopic and electrokinetic studies on how nanoconfined environments modulate CO₂ binding strength, intermediate formation, and reaction kinetics
- **Optimizing reactive capture system to increase energy efficiency:** System-level optimizations on custom BPMs to improve energy efficiency; comparative studies with other systems and preliminary scale-up assessments.

Specific comments:

Comment 1. It is not clear why the authors used 1.5 M carbonate electrolytes. It should be interesting to show the carbonate concentration effect on the catalytic selectivity at identical current density (maybe 200 mA/cm²).

Response: The choice of 1.5 M K₂CO₃ as the feed solution is based on direct air capture (DAC) kinetics. Efficient CO₂ capture from air requires a relatively concentrated KOH solution (~3 M) to achieve reasonable kinetics ([10.1016/j.joule.2018.05.006](https://doi.org/10.1016/j.joule.2018.05.006)). When 3 M KOH is used for DAC, the resulting post-capture liquid is primarily 1.5 M K₂CO₃.

We also investigated the effect of carbonate concentration on electrolysis performance at 200 mA/cm² (**Figure R1**). A K₂CO₃ concentration of 1–1.5 M maintains a high FE_{CO} (>45%). Lower K₂CO₃ concentrations reduce the likelihood of CO₃²⁻ reacting with H⁺ on the BPM surface, leading to insufficient *i*-CO₂ generation and limiting *i*-CO₂ supply to the cathode.

A higher K_2CO_3 concentration may increase $i\text{-CO}_2$ loss during its generation on the CEL surface and its transport to the cathode layer.

Side reaction at the CEL/catholyte interface:

Side reaction during $i\text{-CO}_2$ transport:

Figure R1. Carbonate concentration effect study at 200 mA/cm².

The figure and accompanying discussion have been included as **Supplementary Figure 47** and its corresponding caption.

On page 15 of the manuscript, we now write:

“In terms of DAC capture kinetics (using concentrated KOH) and subsequent reactive capture product selectivity, we selected 1.5 M K₂CO₃ as the feed solution (Supplementary Fig. 47).”

Comment 2. I do not think it is appropriate to call ‘carbonate electrolysis’, because carbonate is not reduced electrochemically, and it is not involved in any electrolysis directly here. It is still CO₂ electrolysis, thereby so-called ‘Carbonate electrolysis’ may mislead the readers.

Response: In the manuscript, we refer to carbonate electrolysis—using carbonate as the aqueous electrolyte—rather than carbonate reduction, as carbonate itself is not directly reduced in the electrolyzer.

To enhance clarity, we now use the term *“in situ CO₂ reduction”* and *“carbonate-liquid-fed electrolyzer”* in the context of electrified reactive capture.

Comment 3. Line 194, it is not clear what ‘single-pass efficiency’ means.

Response: In MEA-based electrolyzers for gas-fed CO₂ reduction, single-pass (SP) efficiency serves as a diagnostic tool to analyze mass transport limitations and catalyst performance (

<https://doi.org/10.1038/s44286-024-00035-3>). This metric provides insights into the catalyst layer's efficiency in utilizing locally supplied CO₂, particularly under electrolysis conditions where CO₂ is catalytically consumed (e.g., CO₂-to-CO conversion). This behavior closely parallels that of carbonate-liquid-fed electrolyzer with ultra-low CO₂ supply rates.

$$\text{Catalytic Single Pass Efficiency (SP): } SP = \frac{\dot{V}_{CO}}{\dot{V}_{CO_2,in}} \times 100\%$$

Where \dot{V}_{CO} and $\dot{V}_{CO_2,in}$ represent the volume flow rates of the produced CO and the total CO₂ inlet, respectively. SP represents the ratio of CO outlet flow to total CO₂ input flow. A high SP indicates efficient CO₂ transfer and catalytic conversion.

Now, we have incorporated this discussion and definition into the **Supplementary Figure 36** caption and **Methods section** of revised manuscript on **page 20** for clarity.

On page 20 of the manuscript, we now write:

“We define the single-pass (SP) efficiency as follows:

$$\text{Single Pass Efficiency (SP): } SP = \frac{\dot{V}_{CO}}{\dot{V}_{CO_2,in}} \times 100\%$$

Where \dot{V}_{CO} and $\dot{V}_{CO_2,in}$ represent the volume flow rates of the produced CO and the total CO₂ inlet, respectively.”

On page 8 of the manuscript, we now write:

*“IsotropicNano achieved an impressive single-pass (SP) efficiency, the ratio of CO outlet flow to total CO₂ input flow, of 45% at 300 mA/cm² (**Supplementary Fig. 36**).”*

Comment 4. The function of the interposer (filter membrane) in the electrolyzer is not clear, and I think it is better to have a clear explanation and description of the importance of the interposer. **Response:** Our previous work ([10.1016/j.joule.2023.05.003](https://doi.org/10.1016/j.joule.2023.05.003)) investigated the function of the interposer. Modeling results showed that the spacing between the cation exchange layer (CEL) of the BPM and the electrocatalyst significantly affects species concentrations in the reactive capture system. When the CEL and catalyst are closely spaced, the locally low pH can promote the hydrogen evolution reaction (HER), while most CO₃²⁻ and OH⁻ diffuse toward the CEL, primarily neutralizing its acidic surface, thereby reducing *i*-CO₂ generation.

Introducing a hydrophilic interposer layer (spacer) between the CEL and catalyst layer (CL) created a pH gradient (**Figure R2**), establishing optimal conditions: a low pH (<4) at the CEL for efficient *i*-CO₂ generation and a relatively higher pH with [CO₂] > 4 vol% at the CL to enhance CO₂RR.

Figure R2. (a) Schematic of cation-exchange layer (CEL), the interposer, catalyst layer (CL), and carbon paper (CP). (b) pH profile of 135 μm spacing at the applied current densities from 200 to 350 $mA\ cm^{-2}$ in 1.5 M of K_2CO_3 . These two figures are adopted from our previous work (*Joule* 7, 1277–1288. 2023).

We have incorporated the discussion on the use of the interposer and included the reference to support it.

On page 13 of the manuscript, we now write:

“A hydrophilic interposer layer was introduced between the CEL and catalyst layer to establish a pH gradient, ensuring a low pH at the CEL for efficient $i-CO_2$ generation and a higher pH at the catalyst layer to facilitate CO_2RR .¹⁴”

Comment 5. I am also wondering if the thickness of the filter membrane (interposer) could correlate with CO_2 generation and transport, which affects the catalytic selectivity.

Response: Both thickness and pore size influence catalytic selectivity. Additional experimental data has been included to investigate these effects. Results indicate that a hydrophilic mixed cellulose ester (MCE) interposer with a thickness of 135 μm and pore size of 8 μm achieved the highest FE_{CO} in carbonate-liquid-fed electrolyzer.

Figure R3. (a) Schematic representation of the BPM-based electrolyzer incorporating an interposer. (b) Optimization of the interposer layer by varying pore size, material type, and thickness at 200 mA/cm^2 in the carbonate feed electrolyzer.

The above figure has been included in **Supplementary Figure 46**.

On page 15 of the manuscript, we now write:

“We optimized the interposer thickness, pore size, and material type, identifying the hydrophilic mixed cellulose ester (MCE) with a 135 μm thickness as achieving the highest FE_{CO} (Supplementary Fig. 46).”

Reviewer #2 (Remarks to the Author):

The manuscript titled “A Permeable Porous Conductive Catalyst Increases the Catalyst:CO₂ Interaction Cross-Section in Electrified Reactive Capture” presents a study on a Ni single-atom catalyst dispersed in a conductive carbon support, forming a permeable porous conductive catalyst (P2C2) intended to enhance CO₂ retention and electroreduction efficiency in an electrified reactive capture system using carbonate capture liquid. While the topic is scientifically relevant and aligns with the scope of the journal, the manuscript suffers from several critical issues that significantly undermine its novelty and scientific contribution. Therefore, I recommend rejecting this submission. The major concerns are outlined as follows:

Response: We sincerely thank the reviewer for their constructive feedback and the opportunity to clarify the novelty, mechanistic insights, and engineering significance of our work. We acknowledge the reviewer’s concerns and have substantially strengthened the revised manuscript to address them. Specifically, we have incorporated a suite of additional experimental and theoretical studies that enhance the scientific depth and highlight the broader relevance of our findings. Key additions include:

1. **Electrokinetic Investigations** – We conducted **electrokinetic studies** to probe CO₂ binding affinity and to determine reaction orders in both diffusion-limited and kinetically controlled regimes. These measurements, along with activation energy analysis, provide deeper insight into the intrinsic catalytic behavior and transport limitations of different porous catalyst structures (See **Figure 4g–h, Supplementary Table 7, Figure S43, and Supplementary Note 5**).
2. **In Situ Spectroscopic Analysis** – Using **in situ ATR-SEIRAS**, we investigated how nanoconfined pore structures influence the stabilization of key intermediates, facilitate CO desorption, and enhance proton activation. These findings elucidate how the interconnected porous architecture contributes to improved CO production (See **Figure 3a–f, Supplementary Table 8, Figure S44**).
3. **Structural Characterization – Synchrotron-based Small-Angle X-ray Scattering (SAXS)** was used to characterize the pore networks in our catalysts. These measurements confirmed the presence of periodically aligned pores, providing a mechanistic link between pore geometry and electrochemical performance (See **Figure 2d, Supplementary Figure 15, and Note 2**).
4. **Further transport investigation** — Through **in situ Electrochemical Mass Spectrometry (EC-MS)**, we directly measured *i*-CO₂ losses at the cathode outlet and identified clear differences between catalysts. Complementary control experiments (varying catalyst loading, pore structure, and interposer design) revealed that full catalyst layer utilization, not just surface activity, is essential for maximizing *i*-CO₂ retention and

conversion in carbonate-fed systems (See Figure 2g, Supplementary Figures 29–31 and 46, and Note 3).

5. **Additional simulation discussions:** We have revised the manuscript to better highlight the originality and impact of our modeling approach. Our hierarchical multiscale framework, combining DFT and MD, reveals how nanoconfinement, interfacial interactions, and pore geometry affect CO₂ adsorption and transport at the molecular level. These insights go beyond traditional continuum models by capturing spatial heterogeneities and informing realistic boundary conditions for future reaction–transport models. We have clarified these points in the revised section, making the novelty and impact of our computational work more accessible to readers. The revised section now clearly emphasizes these contributions.

These integrated investigations provide not only a performance enhancement but also mechanistic and design principles for advancing reactive capture systems. We respectfully submit that, beyond demonstrating improved catalytic performance, our work offers valuable insights into the structure–function–transport relationships within carbonate-fed electrolysis—an emerging and strategically important direction for sustainable carbon capture and utilization (CCU).

Specific comments:

Comment 1. Lack of Novelty: The concept of employing a porous medium catalyst in a carbonate capture liquid electrochemical CO₂ reduction system has already been reported by Lee et al. (Geonhui Lee, Armin Sedighian Rasouli, Byoung-Hoon Lee, Jinqiang Zhang, Da Hye Won, Yurou Celine Xiao, Jonathan P. Edwards, et al., *Joule*, 2023, 7(6): 1277-1288). The primary distinction claimed in this manuscript is the introduction of a Ni single-atom catalyst. However, the authors fail to demonstrate the unique advantages of this catalyst over previously reported systems. There is insufficient discussion and quantitative comparison regarding performance improvements, such as selectivity enhancement, reaction efficiency, or suppression of side products.

Response: We appreciate the reviewer’s insightful comment regarding the novelty of our work. We want to further clarify the novelty and unique contributions of our work in relation to the cited study of our previous work (*Joule*, 2023, 7(6), 1277–1288; DOI: 10.1016/j.joule.2023.05.003).

(1) Distinct product focus and catalytic objectives

Our prior work focused on ethylene (C₂H₄), whereas the current study targets efficient CO/syngas generation. These are fundamentally different objectives with distinct catalytic requirements and system optimizations.

(2) Unique catalyst design and structure-function investigation

The primary goal of our earlier study was to evaluate the effect of the interposer layer on *i*-CO₂ generation and transport. The Cu/CoPc-CNT tandem catalyst used in that work was not designed for optimized porosity nor was its internal structure investigated in detail. In contrast, the current

work introduces a rationally designed Ni-SAC synthesized via a templating method to achieve a 3D interconnected nanoporous structure. This design enables enhanced *i*-CO₂ retention and binding within the catalyst layer, critical for maximizing conversion efficiency under carbonate-fed conditions.

In our current study, we added extra experimental data and systematically investigated how pore morphology, catalyst layer utilization, and active site accessibility govern performance. Through **in situ ATR-SEIRAS**, **electrokinetic analysis**, **in situ EC-MS**, and **synchrotron-based SAXS**, we provide mechanistic insights into CO₂ activation, intermediate stabilization, and CO selectivity enhancement that were not explored in the previous work.

(3) Quantitative performance comparison and system-level benchmarking

To clearly demonstrate the advantages of our approach, we performed a comprehensive literature comparison of reactive capture systems (**Supplementary Tables 9 and 10**, and **Supplementary Note 7**). This includes benchmarking across:

- Different capture media (carbonate, bicarbonate, amines, etc.)
- Target products (CO, formate, C₂H₄, CH₄)
- Faradaic efficiency, energy efficiency, and *i*-CO₂ utilization
- Electrolyzer configuration

These comparisons establish that our system achieves among one of the highest CO selectivity, carbon utilization, and CO concentration reported to date across both reactive capture and gas-fed CO₂ electrolysis systems (acidic, neutral, and basic conditions).

We have revised the manuscript to explicitly highlight these distinctions.

On page 16 of the manuscript, we now write:

“The combination of FE_{CO} , V_{cell} , EE, and stability is comparable to, or exceeds, those reported in previous reactive capture studies utilizing various capture media or targeting different products (Supplementary Table 9).”

On page 16 of the manuscript, we now write:

“This combination of carbon utilization and CO concentration is the highest, looking across both reactive capture and gas-fed CO₂ reduction systems (acidic, neutral, and basic conditions) reported to date (Supplementary Table 10).”

Comment 2. Inadequate Modeling of CO₂ Transport: The section titled “Simulation of *i*-CO₂ transport kinetics in pores” only considers the physical adsorption and diffusion of CO₂ within the pores, without coupling the electrochemical reduction reaction. The dynamic interaction between

CO₂ mass transport and catalytic reaction is a critical aspect in such confined pore systems, and neglecting this coupling renders the simulation results incomplete and of limited practical value. Furthermore, the impact of porous structure on CO₂ transport and adsorption behavior has been extensively studied in prior works, diminishing the originality of this specific modeling approach.

Response: We thank the reviewer for this important comment. We agree that coupling CO₂ transport with electrochemical reactions is important, particularly under confinement. Below, we provide clarification on the objectives and scope of our current work, the technical limitations of existing modeling approaches, and the specific novelty of our simulation framework.

(1) Justification for physisorption focus:

Our study focuses on equilibrium CO₂ physisorption in meso- and nanopores, as it governs the amount of CO₂ available for electrochemical reactions. Physisorption rapidly equilibrates, forming a reservoir that continuously supplies CO₂ for slower, non-equilibrium steps like chemisorption and electrochemical conversion, which require overcoming higher energy barriers. As CO₂ is consumed, rapid gas exchange with the surroundings maintains this physisorbed state. Therefore, the reservoir size directly influences reactant availability and overall performance. This is also supported by electrokinetic data showing that under CO₂-limited conditions, increasing CO₂ partial pressure significantly boosts FE_{CO} (**Supplementary Note 5, Fig. 4g–h**).

(2) Modeling scope and limitations of existing approaches:

Directly modeling the coupling of CO₂ transport with electrochemical reactions at the molecular scale remains extremely challenging due to the inherently multiscale and multicomponent nature of the system. It involves complex interplays between species concentrations, solvent and ion dynamics, interfacial structure, local dielectric properties, and charge transfer kinetics—all of which are difficult to resolve simultaneously with current modeling techniques.

While continuum frameworks such as the Poisson–Nernst–Planck (PNP) equations and Butler–Volmer (BV) kinetics offer valuable mesoscale insights, they rely on several mean-field approximations that are known to break down at nanometer scales. These include assumptions of spatially uniform dielectric constants, continuum ion distributions, and reaction kinetics decoupled from nanoscale confinement and solvation structure. Furthermore, many such models adopt volume-averaged or homogenized pore treatments, which overlook critical molecular-level details due to limited structural resolution or computational constraints.

Given these limitations, our work focuses on disentangling the structural and kinetic effects imposed by pore size and interfacial interactions on CO₂ availability and residence—factors that ultimately govern the reactant supply to catalytic sites under non-equilibrium conditions. By decoupling the physical adsorption and diffusion components, we aim to identify how confinement,

pore geometry, and interfacial interactions influence local CO₂ distribution and retention time—essential metrics for advancing future reactive transport models.

(3) Our insights into nanoconfined transport beyond classical models:

To address the reviewer's concern regarding originality, we emphasize that our simulations—leveraging MD and DFT—go beyond traditional continuum models by resolving confinement-induced deviations in both spatial distribution and transport dynamics of CO₂. Specifically, we: (1) Analyzed CO₂ density profiles as a function of distance from the surface, revealing strong near-wall accumulation driven by short-range non-electrostatic interactions, aligned with DFT-predicted physisorption minima. (2) Applied survival time correlation functions to quantify region-specific CO₂ residence times. We observed that CO₂ in the interfacial region exhibits significantly suppressed mobility compared to the pore center, with slower $S(\tau)$ decay rates especially in sub-2 nm pores (Fig. 3f).

These molecular insights are inaccessible to traditional continuum models, which assume spatially averaged behavior and uniform transport properties. For example, PNP-based models typically neglect short-range non-electrostatic adsorption forces, yet our simulations clearly show enhanced CO₂ layering near the surface due to these interactions—contributing up to fourfold increases in local concentration compared to the bulk (**Fig. 3e**). Moreover, our simulations reveal that ion and molecule transport characteristics are spatially heterogeneous. Specifically, CO₂ mobility differs substantially between the pore interface and central region, and this difference increases with confinement. This is in contrast to conventional models that assign constant diffusion coefficients throughout the domain, thereby overlooking spatial variation in mobility and residence times. These results directly inform the limitations of current reaction–diffusion frameworks and highlight the importance of molecular-level modeling to provide boundary conditions and constitutive inputs for future continuum-scale descriptions.

(4) Technical novelty of our modeling framework:

Our study introduces a hierarchical, multiscale simulation strategy combining first-principles DFT with classical MD, uniquely refined via a genetic algorithm to align force-field predictions with DFT adsorption energies. This refinement ensures that the MD simulations faithfully represent surface-specific interactions, including the enhanced physisorption of CO₂ near atomically dispersed Ni sites versus pristine graphene. Our DFT results show that CO₂ exhibits a physisorption energy minimum of -0.175 eV at a distance of ~ 3.1 Å above the Ni site, with linear geometry retained (180° O–C–O angle), highlighting a weak but favorable interaction. When Ni is removed (i.e., pure graphene), the adsorption energy becomes positive (0.160 eV), further confirming the surface-specific affinity driven by single-atom Ni. These features are encoded into the MD framework through force-field optimization, enabling atomistically accurate simulations of transport behavior in nanoconfined environments.

This combination of rigorously validated DFT-MD integration and nanoconfined CO₂ dynamics represents a significant advancement over prior models that rely on generic parameterizations or neglect confinement effects. By delivering accurate site-specific energetics and dynamic transport metrics, our work offers a quantitative foundation for constructing more physically grounded reaction–transport models in future efforts.

We have revised the manuscript to explicitly articulate these aspects to ensure that the novelty and scope of our modeling approach are clearly conveyed and accessible to readers.

On page 9 of the revised manuscript, we now write:

“Our electrochemical reactive capture systems in confined pores involve tightly coupled processes—transport, adsorption, solvation, charge distribution, and reactions—spanning multiple scales.³³ These effects are strongly influenced by local pore environments and remain challenging to fully capture with current models, especially under nanoconfinement where continuum assumptions break down.

CO₂ adsorption and mobility within nanoporous environments influence reactant availability at electrochemical interfaces.³³ The physisorption of CO₂, in particular, plays a central role by establishing a local reservoir of molecules that are readily available to participate in subsequent, slower steps such as chemisorption and electron transfer.^{34,35} Understanding how pore geometry and surface characteristics influence CO₂ adsorption and diffusion is of use for optimizing catalyst design and performance. In this context, our simulation work focuses specifically on the physisorption and transport behavior of CO₂ in sub-3 nm pores by combining density function theory (DFT) and molecular dynamics (MD) simulations.”

On page 11 of the revised manuscript, we now write:

“Our results offer insights into sub-nanometer variations in CO₂ adsorption density, interfacial residence times, and transport kinetics—key factors in determining reactant availability at active sites. They are complementary to conventional continuum models, such as Poisson–Nernst–Planck or volume-averaged frameworks,³⁷ which can couple ion transport and reaction kinetics but typically assume spatial uniformity and size-independent properties—assumptions that break down under nanoconfinement.³⁸ We hope these insights will inspire the development of future multiscale reaction–transport models—a task that remains both computationally demanding and methodologically underdeveloped.”

Experimentally:

In addition to BET and HRTEM characterization, we incorporated synchrotron-based Small-Angle X-ray Scattering (SAXS) measurements (**Figure R1 and R2**) to gain deeper insights into pore alignment and size distribution. The SAXS results confirmed the presence of porous networks in our templated samples. Among them, *IsotropicNano* displayed the highest surface roughness, the most disordered pore arrangement, the smallest average pore diameter, and the highest nanopore fraction. These results provide important structural parameters that complement other characterization techniques, simulations, and mechanistic studies, contributing to a more comprehensive “structure–performance” relationship.

Figure R1. Small-Angle X-ray Scattering (SAXS) profile.

Figure R2. Fractal analysis in SAXS on (a) DirectionalMeso, (b) IsotropicMeso, and (c) IsotropicNano.

On page 5 of the manuscript, we now write:

“Synchrotron-based Small-Angle X-ray Scattering (SAXS) indicated periodic and symmetric pore arrangements,²⁷⁻²⁹ as shown by correlation peaks in the profiles (Supplementary Note 2 and Fig. 15). Fractal analysis²⁹ showed mass fractal structures (fractal dimensions: $D = 1-3$), where the mass of the porous network scales non-linearly with size, consistent with self-similar structures formed via templating. Higher D values in IsotropicNano suggest increased spatial complexity, indicating greater surface roughness and a more disordered pore structure. The primary SAXS peaks (Fig. 2d), corresponding to different Miller indices, reflect average pore-to-pore distances that decrease from ~15 nm in DirectionalMeso to ~3 nm in IsotropicNano. It indicated IsotropicNano possesses the smallest average pore diameter, consistent with its high nanopore fraction and the cubic lattice arrangement of its pore network (Supplementary Note 2).”

Comment 3. Superficial Data Analysis: Although the authors compare the performance of different porous catalysts, the interpretation of structure-performance relationships is underdeveloped. There is a lack of in-depth analysis of the potential synergistic effects between the microporous structure and the Ni single-atom active sites. For instance, it remains unclear whether the micropores merely serve as CO₂ retention zones or actively modify the electronic structure of the Ni sites. Additionally, the explanation for the observed improvement in Faradaic efficiency is largely qualitative, with limited kinetic data or in situ characterization to substantiate the claims.

Response: We thank the reviewer for highlighting the need for deeper analysis. We fully agree that the deeper analysis of the structure–performance relationship is essential. In response, we have conducted additional electrokinetic measurements and in situ spectroscopic studies to elucidate the role of nanoconfined environments and their interaction with Ni single-atom active sites.

1. Electronic structure studies

XPS characterization (Supplementary Figure 14) show consistent Ni 2p binding energies (~855.2 eV) across all catalysts, indicating a similar oxidation state for the Ni single atoms. N 1s spectra reveal four common nitrogen species—pyridinic N (~398.5 eV), pyrrolic N (~399.8 eV), graphitic N (~400.9 eV), and oxidized N (~402.5 eV)—in comparable ratios. Additional characterization of elemental analysis further showed similar nitrogen content (9-10 wt%, Supplementary Table 2).

These results suggest that the electronic environment of the Ni active sites remains unchanged in the as-synthesized catalysts, and the performance differences among the catalysts primarily stem from variations in pore structure and local mass transport, and the tailored local environment during electrolysis.

2. Electrokinetic studies

To investigate how pore morphology influences CO₂ transport and utilization, we performed detailed electrokinetic studies:

(1) CO₂ binding affinity

Using varying gas CO₂ concentrations in 1.5 M K₂CO₃ electrolyte in three-electrode flow cells, we modeled the CO production rate based on a Michaelis-Menten-type mechanism involving two steps:

1. **CO₂ binding step**, forming an intermediate complex (Ni – CO₂⁻).

2. **Catalytic step**, where the intermediate undergoes conversion to generate the CO product.

The rate expression:

$$j_{\text{CO}} = \frac{nFk_{\text{cat}}^0 K[M^*]_0 P_{\text{CO}_2}}{1 + KP_{\text{CO}_2}} \exp\left(\frac{\beta\eta F}{RT}\right)$$

where $K = \frac{k_a}{k_d + k_{\text{cat}}}$ represents the CO₂ binding affinity constant (details in **Supplementary Note 5**).

IsotropicNano exhibited the highest K value (**Figure R3a**), indicating optimal CO₂ binding affinity, which supports high CO₂RR activity even under CO₂-deficient conditions.

(2) Reaction order analysis

Reaction orders were extracted in both diffusion-limited and kinetically controlled regimes. The diffusion-limited regime for IsotropicNano (0–12% CO₂) was narrower than for DirectionalMeso and IsotropicMeso (up to 20%), indicating **reduced mass transport limitations** in the optimized pore network. In the kinetically controlled region, IsotropicNano showed near-zero reaction order, whereas the other two showed higher values (0.2–0.3), reflecting persistent mass transport dependence (**Figure R3b**).

Figure R3. (a) CO₂ binding affinity study: j_{CO} as a function of partial pressure of CO₂. (b) Reaction order study: $\log j_{\text{CO}}$ as a function of $\log c_{\text{CO}_2}$.

(3) Activation energy

We performed Arrhenius analysis by varying the cell temperature at 2.7 V in the MEA-based cell for CO₂RR. IsotropicNano showed the lowest activation energy (7.6 kJ/mol), compared to 9.2 and 10.4 kJ/mol for the other samples (**Figure R4**), indicating enhanced charge transfer and intermediate conversion efficiency.

Figure R4. Arrhenius plots of CO partial current density as a function of temperature.

3. In situ ATR SEIRAS

We conducted *in situ* attenuated total reflectance–surface-enhanced infrared absorption spectroscopy (ATR-SEIRAS) under simulated low-concentration CO₂ conditions. Controlled potential experiments revealed characteristic peaks for CO₂ reduction intermediates (*COOH, *CO) and interfacial water across all three samples (**Figure R5**).

Figure R5. In situ ATR SEIRAS on three samples.

Figure R6. In situ ATR SEIRAS adsorbed *CO (CO_{atop}) peak (2000–2100 cm^{-1}).

*CO Adsorption and Desorption

The linearly adsorbed *CO (CO_{atop}) peak (2000–2100 cm^{-1}) increased in intensity at more negative potentials (**Figure R6**).^{36,37} IsotropicNano showed ~ 200 mV lower overpotential for *CO appearance compared to DirectionalMeso and IsotropicMeso, indicating enhanced *COOH stabilization or earlier *CO formation. From -0.8 to -1.8 V (vs. Ag/AgCl), IsotropicNano exhibited a blue shift in *CO stretching (2049 \rightarrow 2100 cm^{-1}), suggesting weakened CO binding—likely due to reduced back-donation from metal d-orbitals to CO's π orbital, which favors *CO

desorption and reduces surface poisoning.³⁸ A Stark tuning rate of $\sim 51 \text{ cm}^{-1}/\text{V}$, observed on IsotropicNano, highlights its sensitivity to interfacial electric fields and its facilitation of $^*\text{CO}$ desorption.³⁶ In contrast, $^*\text{CO}$ peaks on DirectionalMeso and IsotropicMeso exhibited minimal Stark tuning or remained unchanged, indicating stronger binding and limited desorption capability.

Hydrogen Bond Network and Water Structure

Figure R7. Deconvolution of O–H stretching peaks.

We also monitored the O–H stretching region ($\sim 3000\text{--}3600 \text{ cm}^{-1}$) to probe hydrogen bonding networks and interfacial water structure (**Figure R7**). All samples showed a red shift of peaks at more negative potentials, indicating increased hydrogen bonding and enhanced water structuring.³⁹ Such structured H-bond networks can stabilize key intermediates ($^*\text{COOH}$ or CO_2^-) through polar interactions, lower the activation barrier for CO_2 reduction, and help sustain a local proton source without promoting excessive HER.⁴⁰

Deconvolution revealed peaks at 3600, 3450, and 3270 cm^{-1} , corresponding to free water, weakly hydrogen-bonded (2-HB), and strongly hydrogen-bonded (4-HB) species, respectively (**Figure R7**).^{41,42} Free water decreased significantly at high overpotentials on all samples. Notably, IsotropicNano showed a higher 4-HB/2-HB ratio and stronger spectral shifts, suggesting more robust H-bond networks that facilitate proton-coupled electron transfer (PCET) and intermediate stabilization during CO_2 reduction.

Conclusion of mechanistic findings

Taken together, these mechanistic studies demonstrate that the nanoconfined 3D pore network in IsotropicNano contributes synergistically to:

- Enhancing CO_2 retention and binding
- Stabilizing reaction intermediates (*COOH, *CO)
- Facilitating CO desorption and minimizing surface poisoning
- Promoting charge transfer and suppressed HER via structured interfacial water
- Reducing mass transport limitations and activation energy

This comprehensive mechanistic understanding substantiates the performance enhancement of IsotropicNano and goes beyond qualitative observations by providing quantitative kinetic and spectroscopic evidence.

We have included the above results and figures into the revised manuscript (**Figure 4a-h**) and **Supplementary Note 5, Figure S43 and S48, and Table S8**.

We have added one subsection in the revised manuscript **page 11-13**, titled “*Nanoconfined environments for enhanced CO_2 binding and activation.*”

On page 11-13 of manuscript, we now write:

“To investigate how $i\text{-CO}_2$ transport within pores influences CO_2 reduction kinetics in a reactive capture system which enhanced local electric field, tailored double layer structure, and the layout or binding angles of CO_2 and H_2O .”

Comment 4. Insufficient Literature Contextualization: The manuscript provides an incomplete review of the state-of-the-art research in carbonate capture liquid-based CO_2 electroreduction.

Recent developments involving Cu-based catalysts and other transition metal single-atom catalysts for selective C₂₊ production in similar systems have not been adequately discussed. This oversight weakens the manuscript's positioning and reduces its clarity in highlighting the research gap it aims to address.

Response: In the revised manuscript, we have added **Supplementary Table 9**, which compares our work with previous studies on aqueous-fed reactive capture systems—including carbonate, bicarbonate, and amine-based approaches—for the production of CO, syngas, formate, and C₂₊ products. The comparison encompasses key performance metrics such as product selectivity, cell voltage, energy efficiency, and overall energy consumption, further underscoring the superior performance and impact of our approach.

We benchmarked carbon utilization and CO concentration against both reactive capture and gas-fed CO₂ reduction systems reported to date, under acidic, neutral, and basic conditions (**Supplementary Table 10**).

Additionally, we calculated the energy consumption in GJ per tonne of syngas produced and compared the reactive capture system with the conventional sequential capture–release process followed by gas-fed CO₂ upgrading via SOEC and RWGS (**Fig. 5d and Supplementary Note 6**).

Finally, we included a discussion in the main text and added **Supplementary Note 7** to compare the differences between carbonate- and bicarbonate-based reactive capture systems.

All of the above comparisons with state-of-the-art studies, along with related discussions, are reflected in the revised manuscript and Supplementary Information.

On page 15-16 of the manuscript, we now write:

*“The combination of FE_{CO} , V_{cell} , EE , and stability is comparable to, or exceeds, those reported in previous reactive capture studies utilizing various capture media or targeting different products (**Supplementary Table 9**).”*

*“This combination of carbon utilization and CO concentration is the highest, looking across both reactive capture and gas-fed CO₂ reduction systems (acidic, neutral, and basic conditions) reported to date (**Supplementary Table 10**).”*

*“We conducted an energy analysis of systems of interest (**Fig. 4d and Supplementary Note 2**).....”*

*“We further identified key differences between carbonate- and previously reported bicarbonate-based reactive capture systems^{21,55} in terms of electrolysis environment, energy consumption, and capital cost (**Supplementary Note 7**).¹⁰ Bicarbonate-liquid-fed electrolyzers typically show high FE_{CO} due to greater i -CO₂ availability and a near-neutral buffered environment that suppresses i -*

CO₂ consumption through local acid-base reactions; however, they exhibit lower i-CO₂ utilization compared to carbonate systems (40–60% vs. >99%).^{21,55} While flue gas capture (4–15% CO₂) via K₂CO₃ (aq) absorbent can produce a bicarbonate-rich solution, it is kinetically unfavorable for DAC due to the low CO₂ concentration (~400 ppm).^{10,11} Converting post-captured carbonate to bicarbonate for continuous electrolysis adds extra processing steps, increasing both capital and operating costs.¹¹ Therefore, improving selectivity and lowering cell voltage in carbonate-based systems is still critical for efficient air-to-product CCU.”

Overall, the manuscript, while addressing a relevant problem, lacks the necessary scientific novelty, rigor in coupling experimental and theoretical analyses, and a comprehensive positioning within the existing literature. Given these substantial limitations, I recommend rejecting the manuscript in its current form.

We sincerely appreciate your time and effort in reviewing our manuscript and providing critical feedback. We acknowledge your concerns regarding scientific novelty, rigor in experimental-theoretical coupling, and positioning within the existing literature. To address these concerns and strengthen our manuscript, we have made significant revisions, including additional experimental and theoretical analyses to reinforce our findings.

Reviewer #3 (Remarks to the Author):

Liu et al. developed several porous carbon materials as supports for nickel single-atom catalysts in a carbonate-fed reactive capture system. The authors also employed the state-of-the-art bipolar membrane and anode material, the system achieved good performance. The design and fabrication of the porous carbon materials appear rational, with improved porosity, the interaction between the catalyst and in-situ CO₂ molecules could be enhanced.

Response: We sincerely appreciate the reviewer's thoughtful evaluation of our work and their recognition of our efforts in designing porous carbon-supported Ni single-atom catalysts, as well as integrating state-of-the-art bipolar membranes and anode materials for carbonate-fed reactive capture systems.

Below, we provide detailed responses to the comments and outline the revisions made to strengthen our manuscript.

I have a few questions for the authors:

Comment 1. The authors mentioned that CO₂ retention is a more important factor compared to nickel content in these catalysts. Did the authors try higher nickel loadings on these carbon materials, and how were the results?

Response: In the revised manuscript (**Supplementary Figure 28, Figure R1** below), we synthesized both DirectionalMeso and IsotropicNano catalysts with varying Ni precursor amounts and quantified the Ni content via ICP-MS. As expected, Ni loading increased with higher precursor concentrations. However, the FE_{CO} showed only marginal improvement or even decreased at higher Ni contents. This decrease in FE_{CO} is likely attributed to the formation of Ni nanoparticles, which tend to favor the HER over CO₂RR.

Figure R1. The FE of product and Ni content in the catalyst as a function of Ni precursor amount.

Additionally, **Supplementary Table 4** compares Ni content across different samples, revealing that IsotropicNano, despite having the lowest Ni loading, achieved the highest FE_{CO} .

These findings suggest that Ni loading alone is not the primary determinant of catalytic activity. Instead, CO_2 transport, and how effectively utilize the Ni active sites play dominant roles in enhancing performance.

We have added the above figure into **Supplementary Figure 28**.

On page 7 of the manuscript, we now write:

“Further increasing the Ni content in IsotropicNano did not enhance performance and may promote nanoparticle formation, which favors the hydrogen evolution reaction (HER) (Supplementary Fig. 28).”

Comment 2. Nickel single atoms have been reported to be selective for CO_2 -to- CO conversion in acidic conditions. Did the authors test the cell without the interposer?

Response: In the revised manuscript, we have included detailed optimizations of the interposer, examining its presence, thickness, pore size, and material type (**Figure R2**). Without an interposer, the FE_{CO} (28%) is significantly lower compared to the configuration with optimized interposer ($FE_{CO} = 47\%$) at 200 mA/cm^2 . The best-performing configuration used a hydrophilic mixed cellulose ester (MCE) interposer, $135\text{ }\mu\text{m}$ in thickness and $8\text{ }\mu\text{m}$ in pore size.

Figure R2. (a) Schematic representation of the BPM-based electrolyzer incorporating an interposer. (b) Optimization of the interposer layer by varying pore size, material type, and thickness at 200 mA/cm^2 in the carbonate feed electrolyzer.

The interposer serves two key functions: **regulating local pH and controlling $i\text{-}CO_2$ availability**. As detailed in our previous work ([10.1016/j.joule.2023.05.003](https://doi.org/10.1016/j.joule.2023.05.003)), the pH profile and $i\text{-}CO_2$ concentration as a function of distance indicate that the spacing between the cation exchange layer (CEL) of the BPM and the electrocatalyst significantly influences species distribution in the reactive capture system.

Without an interposer (closely spaced CEL and catalyst), the locally low pH favors HER. Additionally, CO_3^{2-} and OH^- ions readily diffuse toward the CEL, neutralizing its acidic surface and thereby suppressing *i*- CO_2 generation.

By introducing a hydrophilic interposer layer (spacer), a pH gradient is established:

- Low pH (<4) at the CEL to promote efficient *i*- CO_2 generation
- Higher pH and $[\text{CO}_2] > 4$ vol% near the catalyst layer favors selective CO_2RR

This strategic design effectively balances CO_2 availability and local reaction conditions, improving overall system efficiency.

The above figure has been included in **Supplementary Figure 46** and corresponding discussion was added to the figure caption.

On **page 13** of the revised manuscript, we now write:

*“A hydrophilic interposer layer was introduced between the CEL and catalyst layer to establish a pH gradient, ensuring a low pH at the CEL for efficient *i*- CO_2 generation and a higher pH at the catalyst layer to facilitate CO_2RR .¹⁴ We optimized the interposer thickness, pore size, and material type, identifying the hydrophilic mixed cellulose ester (MCE) with a 135 μm thickness as achieving the highest FE_{CO} (Supplementary Fig. 46).”*

Comment 3. Some studies suggest that nitrogen doping contributes to CO_2 reduction by providing active sites for CO_2 adsorption and conversion, with different nitrogen species exhibiting different activities. The authors used four different nitrogen sources, did this contribute to the differing performances of the final samples?

Response: In the synthesis of all three catalysts, we used the same nitrogen source: ethylenediamine.

Elemental analysis confirmed similar nitrogen content across the samples (9–10 wt%) (**Figure R3**). XPS N 1s spectra revealed four nitrogen species—pyridinic N (~398.5 eV), pyrrolic N (~399.8 eV), graphitic N (~400.9 eV), and oxidized N (~402.5 eV)—present in comparable ratios.

These results indicate that nitrogen content and bonding configurations are consistent among the samples. Therefore, the observed performance differences are more likely attributed to variations in pore structure, catalyst morphology, and CO_2 transport characteristics rather than differences in nitrogen chemistry.

Figure R3. (a) Elemental analysis of N content. (b) XPS N1s results.

We have incorporated the above results into **Supplementary Figure 14 and Table S2**.

On page 5 of the revised manuscript, we now state:

“XPS (Supplementary Fig. 14) showed in each case N 1s spectra of four types of nitrogen (pyridinic, pyrrolic, graphitic, and oxidized N) in similar ratios. Elemental analysis showed similar nitrogen content (9-10 wt%, Supplementary Table 2).”

Additionally, we have included the elemental analysis details and equipment information in the Methods section on page 21.

Reviewer #4 (Remarks to the Author):

In this manuscript, the author presents a catalyst/electrode design that exhibits enhanced performance in the electroreduction of a carbonate solution into a CO+H₂ mixture. Although the performance improvement is notable compared to previous studies, the reviewer finds that the work lacks sufficient novelty, fundamental understanding, and engineering advancement. Consequently, it is not deemed suitable for publication in Nature Communications. It may be appropriate for journals such as EES Catalysis or Chem Catalysis.

Response: We sincerely thank the reviewer for their constructive feedback and the opportunity to clarify the novelty, mechanistic insights, and engineering significance of our work. We acknowledge the reviewer's concerns and have substantially strengthened the revised manuscript to address them. Specifically, we have incorporated a suite of additional experimental and theoretical studies that enhance the scientific depth and highlight the broader relevance of our findings. Key additions include:

- 1. Electrokinetic Investigations** – We conducted **electrokinetic studies** to probe CO₂ binding affinity and to determine reaction orders in both diffusion-limited and kinetically controlled regimes. These measurements, along with activation energy analysis, provide deeper insight into the intrinsic catalytic behavior and transport limitations of different porous catalyst structures (See **Figure 4g–h, Supplementary Table 7, Figure S43, and Supplementary Note 5**).
- 2. In Situ Spectroscopic Analysis** – Using **in situ ATR-SEIRAS**, we investigated how nanoconfined pore structures influence the stabilization of key intermediates, facilitate CO desorption, and enhance proton activation. These findings elucidate how the interconnected porous architecture contributes to improved CO production (See **Figure 3a–f, Supplementary Table 8, Figure S44**).
- 3. Structural Characterization** – **Synchrotron-based Small-Angle X-ray Scattering (SAXS)** was used to characterize the pore networks in our catalysts. These measurements confirmed the presence of periodically aligned pores, providing a mechanistic link between pore geometry and electrochemical performance (See **Figure 2d, Supplementary Figure 15, and Note 2**).
- 4. Further transport investigation** — Through **in situ Electrochemical Mass Spectrometry (EC-MS)**, we directly measured *i*-CO₂ losses at the cathode outlet and identified clear differences between catalysts. Complementary control experiments (varying catalyst loading, pore structure, and interposer design) revealed that full catalyst layer utilization, not just surface activity, is essential for maximizing *i*-CO₂ retention and conversion in carbonate-fed systems (See **Figure 2g, Supplementary Figures 29–31 and 46, and Note 3**).

5. Additional simulation discussions: We have revised the manuscript to better highlight the originality and impact of our modeling approach. Our hierarchical multiscale framework, combining DFT and MD, reveals how nanoconfinement, interfacial interactions, and pore geometry affect CO₂ adsorption and transport at the molecular level. These insights go beyond traditional continuum models by capturing spatial heterogeneities and informing realistic boundary conditions for future reaction–transport models. We have clarified these points in the revised section, making the novelty and impact of our computational work more accessible to readers. The revised section now clearly emphasizes these contributions.

These integrated investigations provide not only performance enhancement but also mechanistic and design principles for advancing reactive capture systems. We respectfully submit that, beyond demonstrating improved catalytic performance, our work offers valuable insights into the structure–function–transport relationships within carbonate-fed electrolysis—an emerging and strategically important direction for sustainable carbon capture and utilization (CCU).

Comment 1. The concept of a “Permeable Porous Conductive Catalyst” is overhyped and actually lacks novelty. Here, “permeable” implies that the gas produced in the liquid phase can penetrate the electrode and be separately collected. However, this phenomenon is common to nearly all catalysts deposited on carbon paper—whether they are nano-sized materials, micro-sized particles, or even sputtered films. Moreover, many studies have already deposited porous, conductive materials—using methods such as templating or dealloying—onto carbon paper and evaluated their performance in CO₂/CO electroreduction. Although these studies focus on the electrolysis of CO₂/CO instead of carbonate, their existence renders the catalyst/electrode design concept presented in this work insufficiently novel.

Response: We thank the reviewer for raising this point. Regarding the "Permeable Porous Conductive Catalyst" (P2C2) concept, we agree that the gas permeability through porous electrodes is indeed a common phenomenon in many catalyst systems deposited on carbon paper, whether composed of nanoparticles, microparticles, or thin films.

To better reflect the focus of our study, we have removed the term "Permeable Porous Conductive Catalyst" from the manuscript title and main text. Instead, we now refer to the catalysts as "DirectionalMeso", "IsotropicMeso", and "IsotropicNano"—terms that explicitly highlight their distinct pore morphologies and size distributions. This adjustment ensures that our discussion centers on the structural properties of the catalysts rather than implying a novel permeability mechanism.

While porous conductive catalysts have been explored in traditional CO₂/CO electroreduction systems, their behavior in carbonate electrolytes **under CO₂-starved, hydrophilic conditions** remain poorly understood. Our work systematically examines how **pore morphology and size distribution** influence mass transport and reaction kinetics in this unique environment,

revealing key differences compared to gas-phase CO₂ reduction systems. By doing so, we provide new design principles for catalysts operating in carbonate media—an aspect that has not been thoroughly investigated in prior studies.

This revised framing more accurately positions our contribution, emphasizing the role of pore structure in overcoming CO₂ transport limitations in carbonate electrolytes rather than overstating permeability as a novel concept.

In the revised manuscript, we have removed "Permeable Porous Conductive Catalyst" from the title, the current title is "*Interconnected Nanoconfining Pore Networks Enhance Catalyst:CO₂ Interaction in Electrified Reactive Capture*".

We have revised the schematic illustration in **Figure 1d** to more accurately represent the conceptual framework of catalyst design in our study.

We also added new experimental data—including electrokinetic analysis, in situ spectroscopy, and control experiments—to investigate how the 3D interconnected morphology and nanopores enhance performance. These results are now included in the manuscript as a new subsection and are presented in the updated **Figure 4**.

Comment 2. A substantial portion of this high-surface-area, porous electrode may not contribute to performance in carbonate electrolysis. The authors suggest that the porous catalyst structure aids in retaining and accumulating CO₂, formed via carbonate acidification, within its pores, thereby enhancing the reduction rate. However, under actual reaction conditions, the by-product OH⁻, generated during CO₂ electroreduction to CO, accumulates significantly within these pores and elevates the local pH. As a result, incoming CO₂ reacts readily with OH⁻—a kinetically more favorable process than the electroreduction on the catalyst surface. This effect is notable even in planar electrodes at the current densities used in the study and is markedly intensified within nanopores. Consequently, only the top layer of the electrode can actively participate in CO₂ reduction, leaving the majority of the active sites inaccessible—a well-documented transport limitation in many electrochemical reactions occurring in porous electrodes.

Response: We thank the reviewer for highlighting this important issue. We conducted additional experimental analyses and discussions on catalyst utilization and penetration depth within porous electrodes under carbonate-fed conditions.

We agree that OH⁻, generated as a by-product during CO₂ electroreduction, can react with *i*-CO₂ under elevated local pH, resulting in acid-base reactions of *i*-CO₂ losses. In our system, the observed FE_{CO} ranged from 20–50%, depending on catalyst type and current density. We quantified the unreacted *i*-CO₂ (g) at the outlet accounted for only ~1–2% in charge efficiency terms, suggesting that a substantial portion of *i*-CO₂ (~50–80%) is lost via OH⁻-CO₂ coupling.

However, despite operating under identical conditions, we observed clear differences in FE_{CO} across different catalysts. These variations are attributed to differences in active site accessibility, CO_2 binding affinity, and $i-CO_2$ retention efficiency within the porous structures. To investigate this further and address the reviewer’s concerns, we performed additional control experiments outlined below.

Figure R1. Comparison between gas-fed CO₂RR and liquid-fed carbonate systems, and between porous single-atom catalysts and non-porous Ag nanoparticles.

(1) Gas-fed vs. carbonate-fed CO₂ electrolysis

The fundamental difference between gas-fed CO₂RR and liquid-fed carbonate electrolysis lies in reactant availability and catalyst environment. Gas-fed systems benefit from a hydrophobic gas-diffusion layer that ensures a sufficient CO₂ supply, whereas liquid-fed carbonate systems operate in a hydrophilic, fully flooded environment where $i-CO_2$ (g) (aq) (in both gas and liquid form) is limited and must be efficiently utilized.

We compared **gas-fed and carbonate-fed** systems using MEA-based setups with the same Ni-SAC catalyst.

- In gas-fed systems, with pure CO₂ gas, FE_{CO} remained >95% across a wide catalyst loading range (0.5–2 mg/cm², **Figure R1a**).
- When gas CO₂ partial pressure was lowered to 20%, catalyst loading began to impact FE_{CO} (**Figure R1b**).

- Under carbonate-fed conditions (200 mA/cm^2), catalyst loading had a much stronger influence: reducing loading from 2 to 0.5 mg/cm^2 resulted in a 50% FE_{CO} drop (**Figure R1c**).

This suggests that in the fully flooded, CO_2 -limited carbonate system, the full depth of the catalyst layer contributes to activity, not just the top surface. Efficient utilization of $i\text{-CO}_2$ depends on pore structure and transport properties that enable $i\text{-CO}_2$ to access internal active sites.

(2) Porous single-atom vs. non-porous nanoparticle catalysts

We compared porous Ni-SAC and non-porous Ag nanoparticles in carbonate-fed systems:

- Ni-SAC, with $\sim 1\text{--}2 \text{ wt}\%$ Ni dispersed in a porous matrix, exhibited a strong dependence on catalyst loading and thickness (**Figure R1c**).
- Ag nanoparticles, with dense active site packing, showed much weaker sensitivity to loading changes (**Figure R1d**).

These findings suggest that porous single-atom catalysts require optimized loading and structure to ensure $i\text{-CO}_2$ reaches all active sites within the porous matrix. The lower active site density in Ni-SAC amplifies the impact of catalyst loading and active site utilization.

(3) Catalyst thickness control

Figure R2. Ni-SAC catalyst thickness effect in carbonate system.

To isolate the effect of catalyst layer thickness, we maintained a constant total loading while varying the Ni-SAC:carbon ratio (3:1 to 1:3, **Figure R2**).

Despite similar total thickness, FE_{CO} strongly correlated with Ni-SAC thickness/content, indicating that pore accessibility and internal site utilization, not just surface exposure, govern performance in carbonate-fed systems.

(4) Local OH⁻ buffering via high flow rate

Figure R3. Effect of flow rate in carbonate system.

We used high electrolyte flow rates (>20 mL/min) in the liquid-fed system to maintain relatively stable local pH and suppress OH⁻ accumulation.

- Both K₂CO₃ and K₂CO₃/KHCO₃ buffer systems benefited from high flow, which reduced local OH⁻ concentration and partially mitigated OH⁻–CO₂ coupling (**Figure R3**).
- This strategy supports deeper catalyst layer utilization by maintaining CO₂ availability throughout the porous electrode.

(5) In situ EC-MS: direct measurement of *i*-CO₂ loss

Figure R4. EC-MS analysis in carbonate system at different current density.

We used in situ EC-MS to detect unreacted $i\text{-CO}_2$ (g) at the cathode outlet (**Figure R4**):

- The IsotropicNano catalyst showed minimal CO_2 (g) evolution at high current densities.
- DirectionalMeso, with its one-dimensional diffusion paths, resulting in faster $i\text{-CO}_2$ escape and showed poorer CO selectivity under the same conditions.
- The 3D interconnected pore network in IsotropicNano, with a higher nanopore fraction, enhanced $i\text{-CO}_2$ retention and promoted utilization throughout the catalyst layer.

These results demonstrate that **entire-layer utilization**—enabled by tailored pore structure and diffusion pathways—plays a key role in carbonate-fed electrolysis.

Summary and manuscript updates:

Our results answer the reviewer’s concern: while OH^- - CO_2 reactions do occur, the data clearly support that **not only the top layer, but the entire optimized porous catalyst layer** contributes to performance when optimized for structure and mass transport in the carbonate-liquid-fed system. Catalyst design, including interconnected nanopores, are critical to overcoming $i\text{-CO}_2$ limitations.

These new results and discussions have been added to **Figure 2g, Supplementary Note 3 and Figures 29–31**.

On page 7 of the revised manuscript, we now state:

*“Control experiments showed that, unlike hydrophobic gas-fed CO_2 RR systems or non-porous Ag-based catalysts—where only the surface layer actively participates in the reaction—an optimal IsotropicNano loading of 1.5–2 mg/cm^2 is essential in the fully-flooded carbonate system to maximize Ni single-site utilization throughout the entire catalyst layer (**Supplementary Note 3 and Figures 29-31**).*

*We further employed in situ electrochemical mass spectrometry (EC-MS) to detect the unreacted $i\text{-CO}_2$ (g) at the cathode outlet (**Fig. 2g**). IsotropicNano showed minimized CO_2 (g) present in the tailgas, especially at high current densities. In contrast, the one-dimensional/directional diffusion in DirectionalMeso may provide too-efficient egress of $i\text{-CO}_2$ from the entrance of the porous catalyst layer (facing the interposer and the CEM side of the BPM) to its plane of egress (atop the GDL), preventing selectivity in $i\text{-CO}_2$ RR from being retained at high currents. The 3D interconnected pore network in IsotropicNano—particularly with its higher fraction of nanopores—enhanced $i\text{-CO}_2$ retention and promoted its efficient utilization within the catalyst layer.*

These results highlight that the entire catalyst layer plays a crucial role in facilitating $i\text{-CO}_2$ transport, retention, and penetration into the interconnected pores, thereby maximizing catalytic efficiency.

Comment 3. In fact, many previous studies on CO₂ electroreduction to formate represent a significantly more efficient approach for converting carbonate into value-added products. Whether using an H-type cell or a microfluidic flow cell with catalysts such as Sn, Pb, In, or Bi-based materials, as long as a bicarbonate solution—readily obtained by saturating a carbonate solution with CO₂—is employed as the electrolyte, the process essentially converts bicarbonate into formate. This method achieves much higher Faradaic efficiencies and current densities. Previous researchers simply did not market these studies as efficient conversions of alkaline-captured CO₂. Although syngas may be more widely demanded than formate, from a scientific standpoint, the concept in the current manuscript does not contain sufficient novelty.

Response: We have added more discussions to clarify the novelty and distinction of our work.

(1) Distinction between gas-fed CO₂ electrolysis and liquid-carbonate-fed reactive capture:

While many previous studies on CO₂ reduction to formate (using Sn, Pb, In, or Bi catalysts) employed bicarbonate-containing electrolytes in H-type or flow cells, these systems typically relied on gas-fed pure CO₂ gas as the carbon source. In such cases:

- Pure CO₂ gas is continuously bubbled into the electrolyte, and bicarbonate serves primarily as a **supporting electrolyte**, not the carbon source (e.g., *ACS Appl. Mater. Interfaces* 2022, 14, 14210–14217; *Energy Environ. Sci.* 2021, 14, 4998–5008).
- Catalysts operated in hydrophobic environments using gas diffusion layers (GDLs).
- These systems require energy-intensive CO₂ regeneration, circulation, and downstream separation after CO₂ capture.

In contrast, our work uses a liquid-fed carbonate electrolyzer, where:

- **Carbonate serves directly as both the carbon source and the electrolyte**, eliminating the need for gas-phase CO₂ input.
- *i*-CO₂ (g) (aq) (in both liquid and gas form) is electrochemically generated at the BPM interface and immediately reduced within a **hydrophilic catalyst environment**.
- This **integrated reactive capture** approach avoids the energy- and cost-intensive CO₂ regeneration and gas handling steps required in traditional processes.

Importantly, the carbonate-based reactive capture system cannot be implemented in H-type or conventional flow cells; it requires a MEA to facilitate localized *i*-CO₂ generation at the membrane–catalyst interface.

(2) Scientific challenges in carbonate-fed electrolyzers vs. gas-fed systems

Compared to gas-fed systems, carbonate-fed electrolyzers present fundamentally different and more complex challenges:

- They operate under *i*-CO₂-limited conditions, requiring catalysts with high intrinsic activity and optimized *i*-CO₂ adsorption/retention capabilities.
- The hydrophilic environment increases the probability of parasitic OH⁻-CO₂ coupling, necessitating careful pore structure design and membrane configuration.
- These unique constraints demand new catalyst and device strategies, such as the 3D interconnected nanoporous catalyst layer demonstrated in our work.

To address this and related concerns, we have revised the manuscript to highlight the novelty and impact of our carbonate-based reactive capture system:

A comparative analysis (**Figure 5d, Supplementary Table 10**) now contrasts our system with traditional sequential CO₂ capture–regeneration–electrolysis workflows, including gas-fed CO₂RR, SOEC, RWGS pathways.

On page 16 of the MS, we now write:

*“This combination of carbon utilization and CO concentration is the highest, looking across both reactive capture and gas-fed CO₂ reduction systems (acidic, neutral, and basic conditions) reported to date (**Supplementary Table 10**).”*

*“We conducted an energy analysis of systems of interest (**Fig. 5d and Supplementary Note 6**): i) sequential capture-and-release followed by gas-fed electrochemical CO₂ upgrade; ii) sequential capture-and-release followed by reverse water-gas shift (RWGS) using hydrogen from an efficient water electrolyzer; iii) the integrated reactive capture approach explored in this study and prior studies, with H₂ is supplemented from a water electrolyzer to obtain H₂:CO of 2:1 syngas. The integrated reactive capture process in this work offers the lowest projected energy expense in GJ/tonne of syngas (43 GJ/tonne), its advantage deriving the avoidance of the CO₂ regeneration and circulation steps in sequential DAC-plus-electrolysis.”*

We expanded our comparison with previous reactive capture systems, covering:

- Carbamate- and KHCO₃-based systems (for flue gas capture using amine and K₂CO₃, 4–15% CO₂).
- K₂CO₃-based systems (for DAC using KOH, ~400 ppm CO₂).
- Target products (CO, syngas, formate, CH₄, ethylene).
- Performance metrics: FE, cell voltage, energy efficiency, and carbon utilization.

On page 16 of the manuscript, we now write:

*“The combination of FE_{CO}, V_{cell}, EE, and stability is comparable to, or exceeds, those reported in previous reactive capture studies utilizing various capture media or targeting different products (**Supplementary Table 9**).”*

“We further identified key differences between carbonate- and previously reported bicarbonate-based reactive capture systems^{21,55} in terms of electrolysis environment, energy consumption, and capital cost (Supplementary Note 7).¹⁰ Bicarbonate-liquid-fed electrolyzers typically show high FE_{CO} due to greater $i\text{-CO}_2$ availability and a near-neutral buffered environment that suppresses $i\text{-CO}_2$ consumption through local acid-base reactions; however, they exhibit lower $i\text{-CO}_2$ utilization compared to carbonate systems (40–60% vs. >99%).^{21,55} While flue gas capture (4–15% CO_2) via K_2CO_3 (aq) absorbent can produce a bicarbonate-rich solution, it is kinetically unfavorable for DAC due to the low CO_2 concentration (~400 ppm).^{10,11} Converting post-captured carbonate to bicarbonate for continuous electrolysis adds extra processing steps, increasing both capital and operating costs.¹¹ Therefore, improving selectivity and lowering cell voltage in carbonate-based systems is still critical for efficient air-to-product CCU.”

Response to Reviewer's Comments

Title: Interconnected Nanoconfining Pore Networks Enhance Catalyst:CO₂ Interaction in Electrified Reactive Capture

Manuscript ID: NCOMMS-25-07597A-Z

We thank the reviewers for their deep engagement with this work and their expert advice. We have acted on each comment and revised the work.

A point-by-point response to the comments is provided below. In the revised manuscript, we use yellow text to indicate additions and changes made in light of referee advice.

Reviewers' Comments:

Reviewer #1 (Remarks to the Author):

The authors have addressed my concerns properly, and I have no more comments.

Reviewer #2 (Remarks to the Author):

All the concerns raised have been adequately addressed.

Reviewer #3 (Remarks to the Author):

Thanks for the responses. The result of nickel loading study is helpful. The authors mention increasing Ni content may promote nanoparticles formation, did the authors observe this through TEM imaging?

Response: We appreciate the reviewer's follow-up comment regarding Ni content. In the revised manuscript, we have included additional TEM images (**Figure R1**) of high-Ni-loading samples (3.3 wt% of Ni). In the as-synthesized state, Ni nanoparticles (visible as white dots) are clearly observed. After electrolysis at 300 mA/cm² for 4 hours, the proportion of Ni nanoparticles further increases.

XRD analysis (**Figure R2**) of the as-synthesized high-loading sample revealed diffraction peaks at ~44.5°, 51.8°, and 76.4° (corresponding to the (111), (200), and (220) planes of fcc Ni, PDF# 04-0850) corresponding to metallic Ni nanoparticles, indicating the presence of Ni particulates in addition to atomically dispersed Ni species.

These results suggest that higher Ni loading leads to nanoparticle aggregation during synthesis and electrolysis, which may enhance hydrogen evolution and suppress *i*-CO₂-to-CO conversion activity in carbonate-fed electrolysis systems.

These results further support our claim that the design of the porous structure—specifically, maximizing active site accessibility—is critical for enhancing mass transport and improving reaction activity in carbonate-fed systems.

We have now added this HRTEM image and XRD to **Supplementary Figure 28-29**.

Before electrolysis

After electrolysis

Figure R1. HRTEM images of high-Ni-loading samples before and after electrolysis. The electrolysis was performed at 300 mA cm⁻² for 4 hours.

Figure R2. XRD on isotropicNano with 100 mg Ni precursor.

In the revised manuscript page 3, we now state:

“Further increasing the Ni content in IsotropicNano did not enhance performance: it leads to nanoparticle formation, favoring the hydrogen evolution reaction (HER) (Supplementary Fig. 28-29).”

Reviewer #4 (Remarks to the Author):

The authors provided an unnecessarily lengthy response yet failed to directly address this reviewer's straightforward comments, ultimately failing to persuade.

Response: We now provide a more succinct summary of actions in response.

Specific comments:

Comment 1. In their response to comment 1, the authors acknowledged that their electrode structure is already common in CO₂ electrochemistry. They then shifted their argument to highlight the conductive porous networks of the catalyst. However, they did not provide convincing evidence that these convoluted nanopores actually enhance the reduction reaction.

Response: We now present more clearly in the revised manuscript how carbonate-fed electrolysis, in which CO₂ (including gas and aqueous CO₂) must be generated *in situ* from CO₃²⁻, is enhanced using nanoporous catalysts. The studies of the system include:

- (1) **Mass transport and CO₂ retention:** EC-MS, EIS (capacitance and relaxation time), CO₂ adsorption isotherms, and low-concentration CO₂ electrolysis experiments reveal enhanced *i*-CO₂ retention and transport in nanoporous catalysts.
- (2) **Computational modeling:** DFT and MD simulations to study how confined nanopore environments affect *i*-CO₂ transport dynamics and local enrichment.
- (3) **Additional mechanistic probing:** In situ ATR-SEIRAS and electrokinetic analysis reveal stronger *i*-CO₂ adsorption and improved CO intermediate formation in nanoporous structures.
- (4) **Structural characterization:** HRTEM, HAADF-STEM/EDS, BET (meso- and nanopore analysis), XRD, and XPS to report on the pore morphology.

We have sought to improve presentation of the work, especially in the manuscript text covering Figures 2–4.

Comment 2. Particularly in their response to comment 2, the authors admitted that approximately 50-80% of the CO₂ reacts with OH⁻ generated during surface reduction, strongly suggesting these pores are inaccessible to CO₂ from the electrolyte—a well-known limitation in electrolysis involving gaseous reactants with restricted mass transfer.

Response: In the revised work we now better explain that *i*-CO₂ at the outlet accounts for ~1–2% of the charge efficiency, from which we conclude that the dominant reactions occurring in the system are ***i*-CO₂-to-CO electroreduction and OH⁻-CO₂ neutralization (98-99% charge efficiency in total)**. We write on page 7 of the revised main text:

“It was only when we combined multidirectional diffusion with the addition of < 2 nm nanopores in IsotropicNano that FE_{CO} rose to 50%±3%, accompanied by the lowest extent of the OH⁻-CO₂

neutralization side reaction and the lowest CO₂ gas evolution at the cathode outlet (Supplementary Table 4).”

Comment 3. In response to comment 3, when comparing their system to a CO₂-bicarbonate system, the authors inaccurately stated, "These systems require energy-intensive CO₂ regeneration, circulation, and downstream separation after CO₂ capture." This claim is fundamentally incorrect because: (1) CO₂ regeneration is not required in such systems, and CO₂ is not consumed significantly, thus eliminating substantial sourcing costs; and (2) circulating CO₂ at atmospheric pressure is not energy-intensive.

Figure R1. Sequential process of direct air capture (DAC) coupled with CO₂ electrolysis (Route 1) vs. integrated process of reactive CO₂ capture (Route 2).

Response: We resolve this concern by becoming quantitative: we now discuss in the revised text an estimated ~8-10 GJ/tonCO₂ for the regeneration step, and an estimated ~ 2 GJ/tonCO₂ for the recirculation step. This is now more clearly presented in the text discussing Figures 1a-b and 5d, and Supplementary Table 11 and Note 6.

In summary, the authors have overstated concepts that are neither fundamentally novel nor practically impactful, and their results remain mediocre.

The revised manuscript now more clearly highlights the diffusion and transport differences between conventional gas-fed CO₂ electrolysis and the reactive carbon capture system studied here.